# DATA-EFFICIENT TRAINING BY EVOLVED SAMPLING

## ABSTRACT

Data selection is designed to accelerate learning with preserved performance. To achieve this, a fundamental thought is to identify informative data samples with significant contributions to the training. In this work, we propose **Evolved Sampling** (**ES**), a simple yet effective framework for *dynamic* sampling along the training process. This method conducts *batch* level data selection based on the dynamics of losses and augmented *loss differences*, which enables flexible *frequency tuning*, and hence significantly reduces the back propagation time with maintained model performance. Due to its conciseness, ES is also readily extensible to incorporate *set* level data selection (to form ES with pruning, **ESWP**) for further accelerations. As a plug-and-play framework, ES(WP) consistently achieves lossless training accelerations across various pre-training and post-training tasks, saving up to nearly 45% wall-clock time. Our results motivate further investigations on the data efficiency aspect of modern large-scale machine learning.

## 1 INTRODUCTION

Deep learning has showcased remarkable performance across a variety of real-world applications, particularly leading to unparalleled successes of large "foundation" models (Touvron et al. (2023); Rombach et al. (2022)). On the other hand, since these large models are usually trained on web-scale datasets, the overall computation and memory loads are considerably increasing and unsustainable, calling for more *efficient* developments of modern large-scale machine learning.

Efficient learning involves several aspects, centering around models, data, optimization, systems, and so on (Shen et al. (2023)). For *data*-efficient machine learning, the core is to properly evaluate the importance per data sample in the original (large) datasets. A broad array of methods is applied in a *static* manner, or known as the offline (coreset) selection, where the samples' importance is determined before the formal training. By leveraging feature representations of data (Swayamdipta et al. (2020); Xie et al. (2023b)), this importance can be either evaluated based on a variety of metrics such as distances (Huang et al. (2023); Xia et al. (2023); Abbas et al. (2023)), uncertainties (Coleman et al. (2020); Margatina et al. (2021)), errors (Toneva et al. (2019); Paul et al. (2021)), etc., or learned via procedures from the meta optimization (Killamsetty et al. (2021c;b); Jain et al. (2024); Wang et al. (2022)) and dataset distillation (Nguyen et al. (2021); Wang et al. (2022); Zhao & Bilen (2023)), or directly assessed by LLMs (Sachdeva et al. (2024)). See more detailed discussions in Appendix Sec. A. However, these approaches can be prohibitively expensive to apply in practice, since their potential dependence on feature representations requires additional (pre-)training in advance.

Another array of methods lies in a *dynamic* sense, or known as the online (batch) selection, where the samples' importance is simultaneously evaluated along the training process. Dynamic sampling methods can be further divided into two categories: *set* level selection, to prune the whole dataset at the beginning of each epoch (Qin et al. (2024); Raju et al. (2021); Thao Nguyen et al. (2023); Attendu & Corbeil (2023)), and *batch* level selection, to sample subsets from original batches for back propagation (Kawaguchi & Lu (2020); Katharopoulos & Fleuret (2017; 2018); Mindermann et al. (2022)). Nevertheless, these dynamic sampling methods leverage similar strategies to evaluate the samples' importance. Based on the naive intuition that samples' contributions to the learning are directly associated with gradient updates, it is natural to re-weight data samples with scales of gradients or losses during training. Sampling methods based on the gradients (Hanchi et al. (2022); Wang et al. (2024b); Gu et al. (2025); Wang et al. (2025; 2024a)) usually suffer from significant computation and memory loads. Sampling methods based on the loss dynamics can involve current losses (Jiang et al. (2019); Loshchilov & Hutter (2016); Qin et al. (2024); Thao Nguyen et al. (2023);

Kumar et al. (2023); Balaban et al. (2023)) and historical losses (Attendu & Corbeil (2023); Raju et al. (2021); Sagawa et al. (2020)) and also adopt reference models (Mindermann et al. (2022); Deng et al. (2023); Xie et al. (2023a)). See more detailed discussions in Appx. A. However, these approaches exploit the information of losses inadequately by only involving "absolute" loss values, without finer considerations on their dynamical "variations" during training.

To tackle these issues, we propose a simple novel dynamic sampling framework, **Evolved Sampling** (**ES**). Unlike previous sampling methods, ES determines the importance/weights of data samples based on both (zero-order) losses and additional (first-order) loss *differences* along the training dynamics. By augmenting and balancing these two orders, ES can **flexibly tune the portion of oscillations (high frequencies) presented in loss signals**, and conducts *batch* level selection without the demand of pre-trained reference models. Importantly, ES employs an equivalent dynamical scheme to compute sampling weights *without explicitly storing historical losses*, and *only computations regarding losses are involved* to *implicitly calculate the required loss differences*, implying the negligible memory costs and mild computation overhead additionally introduced by weight calculations. Due to its simplicity, ES is effortless to implement, while significantly reducing the number of samples used for back propagations (BPs) and consequently saving the overall wall-clock time without degrading the overall performance. Moreover, ES facilitates convenient extensions to data pruning on the *set* level, i.e., **Evolved Sampling with Pruning** (**ESWP**), leading to further accelerations with lossless learning performance. We demonstrate the differences in details between our proposed methods (ES/ESWP) and previous dynamic sampling methods in Tab. 1.

Table 1: Comparison of different dynamic sampling methods. The "history" denotes whether the method uses historical (loss) information along the training. The "dif" column stands for whether the method uses dynamical variations of losses during the training. The last column summarizes the ratio of samples used for back propagations (BPs) relative to the standard training. Here, $r$ stands for the pruning ratio for *set* level methods (pruning data samples of the whole epoch), and $b/B$ represents the pruning ratio for *batch* level methods (selecting a mini-batch $\mathfrak{b}$ (subset) from a meta-batch $\mathcal{B}$).

| | set | batch | history | dif | pct. of samples for BP |
|---|---|---|---|---|---|
| UCB (Raju et al. (2021)) | ✓ | | ✓ | | $1 - r$ |
| KA (Thao Nguyen et al. (2023)) | ✓ | | | | $1 - r$ |
| InfoBatch (Qin et al. (2024)) | ✓ | | | | $1 - r$ |
| Loss (Katharopoulos & Fleuret (2017)) | | ✓ | | | $b/B$ |
| Order (Kawaguchi & Lu (2020)) | | ✓ | | | $b/B$ |
| **ES (ours)** | | ✓ | ✓ | ✓ | $b/B$ |
| **ESWP (ours)** | ✓ | ✓ | ✓ | ✓ | $(1 - r)b/B$ |

Our contributions can be summarized as follows:

- On the theoretical side, we provide quantitative convergence analysis of the loss re-weighted gradient descent (GD) under idealized settings. Motivated by this, we propose a simple novel dynamic sampling framework ES(WP) that can *implicitly* incorporate (and balance) additional dynamical *differences* of losses *without explicitly storing historical values and calculating variations*. By further injecting higher-order dynamical information, one can flexibly tune the portion of oscillations (high frequencies) presented in loss signals with quantitative guidance.

- On the empirical side, we carry out extensive experiments to verify the effectiveness, efficiency, and flexibility of ES(WP). It is shown that ES(WP) consistently achieves lossless training accelerations across various pre-training and post-training tasks, saving up to 45% training time.

The rest of this paper is organized as follows. In Sec. 2, we provide the motivation of loss-based dynamic sampling methods. In Sec. 3, we present the proposed methods with theoretical justifications and complexity analysis. Experiments and ablation studies are provided in Sec. 4. The discussions and outlook are provided in Sec. 5. Related works and all the details of proofs and experiments are in the appendices.

**Notations.** We use normal letters to denote scalars, and boldfaced lower-case letters for vectors. We denote the cardinality of a set $S$ by $|S|$. Let $[n] := \{1, 2, \ldots, n\}$ for $n \in \mathbb{N}_+$. Let $\mathbf{1}_n \in \mathbb{R}^n$ be the vector of all ones. $\lceil c \rceil$ represents the smallest positive integer such that $\lceil c \rceil \geq c$. We use the big-O notation $f(t) = O(g(t))$ to represent that $f$ is bounded above by $g$ asymptotically, i.e., there exists a universal $c > 0, t_0 > 0$ such that $f(t) \leq cg(t)$ for any $t \geq t_0$.

## 2 PRELIMINARIES AND MOTIVATIONS

### 2.1 PRELIMINARIES

The classic setting of general machine learning tasks is as follows. Given a dataset $\mathcal{D} := \{z_i\}_{i=1}^n$ with $z_i := (x_i, y_i)$ (labeled) or $z_i := x_i$ (unlabeled) of the size $n \in \mathbb{N}_+$, the goal is to solve the empirical risk minimization (ERM) problem: $\min_{\theta \in \Theta} \hat{L}_n(\theta) := \frac{1}{n} \sum_{i=1}^n \ell_i(\theta)$, where $\ell_i(\theta) := \ell(f(x_i; \theta), y_i)$ or $\ell_i(\theta) := \ell(f(x_i; \theta))$. Here, $\ell(\cdot, \cdot)$ or $\ell(\cdot)$ denotes the non-negative loss function, and $\hat{L}_n(\theta)$ represents the empirical averaged loss over $n$ data samples. When $n$ is large, a common routine is to compute stochastic gradient on a random batch instead of the whole training set. For instance, starting from an initialization $\theta(0) = \theta_0$, the SGD optimizer updates model parameters by $\theta(t+1) = \theta(t) - \eta_t \nabla_\theta \hat{L}_n(\theta(t)) \approx \theta(t) - \frac{\eta_t}{B} \sum_{j \in \mathcal{B}_t} \nabla_\theta \ell_j(\theta(t))$, where $\{\eta_t\}_{t \in \mathbb{N}}$ denotes learning rates, and $\mathcal{B}_t \subset [n]$ denotes the batch with the size $|\mathcal{B}_t| = B \leq n$. The standard sampling method is to draw the batch $\{z_{i_j}\}_{j=1}^B \subset \mathcal{D}$ uniformly without replacement for $\lceil n/B \rceil$ iterations in one epoch, which we refer as the standard batched sampling (baseline, no data selection).

### 2.2 THEORETICAL MOTIVATIONS

Obviously, the standard batched sampling takes equal treatment to data samples. This can be *inefficient* since different samples may have varied importance to the learning task at different training stages: As the training proceeds, there are inevitably samples that are fitted more accurately compared with the others, leading to lower priority to learn these better-fitted samples in the sequel. Hence, it is necessary to assign *adaptive* weights for data samples during training.

**Convergence of loss re-weighted GD.** As discussed before, it is intuitively reasonable to measure the samples' importance with scales of losses along the training, putting more weights on samples with larger losses. The experiments in Katharopoulos & Fleuret (2017) and Kawaguchi & Lu (2020) have suggested that this kind of "loss-weighted" gradient decent dynamics can accelerate learning in practice compared to vanilla GD (without data re-weighting). To step further, this work develops these former literatures in theory by first mathematically proving the following convergence rate.

**Proposition 2.1** (Reduced version; see a full version in Prop. B.1). *Consider the continuous-time idealization of the loss-weighted gradient decent, i.e.*

$$\frac{\mathrm{d}}{\mathrm{d}s} \hat{\theta}_n^{lw}(s) = -\sum_{i=1}^n \frac{\ell_i(\hat{\theta}_n^{lw}(s))}{\sum_{j=1}^n \ell_j(\hat{\theta}_n^{lw}(s))} \nabla_\theta \ell_i(\hat{\theta}_n^{lw}(s)), \tag{2.1}$$

*with the initialization $\hat{\theta}_n^{lw}(0) = \theta_0$. Assume that there exists $\theta^*$ such that $\hat{L}_n(\theta^*) = 0$ and $\ell_i(\cdot)$ is convex for each $i \in [n]$. Then, we have the more-than sub-linear convergence rate of Eq. (2.1), i.e., there exists $s_0 \in [0, s]$ such that*

$$\hat{L}_n(\hat{\theta}_n^{lw}(s_0)) - \hat{L}_n(\theta^*) \leq \frac{1}{2s} \|\theta_0 - \theta^*\|_2^2 - \frac{1}{s} \int_0^s \Delta(s') \mathrm{d}s', \quad s > 0, \tag{2.2}$$

*where $\Delta(\cdot)$ is a positive-valued function on $[0, \infty)$.*

Prop. 2.1 suggests that (under certain regularity conditions) the loss-weighted gradient flow converges more than sub-linearly to the global minimum, while the standard gradient flow (i.e. the continuous-time idealization of vanilla GD) only has the sub-linear convergence.[1]

To formulate, for any $i \in [n]$ and $t \in \mathbb{N}$, define $w_i(t)$ as the (unnormalized) weight of the $i$-th sample at the $t$-th (training) step. For the standard batched sampling, we obviously have the uniform weights: $w_i(t) \equiv 1/n$. For the loss-weighted sampling Eq. (2.1), one calculates the sampling probability as

$$p_i(t) \propto w_i(t) = \ell_i(\theta(t)), \tag{2.3}$$

i.e., the weight is set as the current (non-negative) loss value. On top of that, there are also some variants of loss-weighted sampling strategies: For instance, Kumar et al. (2023) sets $w_i(t) = g(\ell_i(\theta(t)))$, where the function $g(\cdot)$ is pre-defined based on the theory of robust optimization; Kawaguchi & Lu (2020) directly selects top-$q$ samples in terms of current losses per training step, which can be regarded as another realization of Kumar et al. (2023).

---

[1]Although this sharper convergence bound cannot imply learning accelerations solely in theory, accelerations are often observed in practical simulations (e.g. Table 1, 3 and Figure 3, 4 in Kawaguchi & Lu (2020)).

# 3 METHODS AND ANALYSIS

## 3.1 EVOLVED SAMPLING

In general machine learning tasks, the typical behaviors of averaged losses often appear decent trends overall, but can oscillate meanwhile due to the noises in training dynamics. This introduces the instability issue of sampling schemes (e.g. Eq. (2.3)) applied in practice, i.e., the loss-weighted sampling scheme like Eq. (2.3) is intrinsically *sensitive* to possibly large *variations* of (individual) losses and not robust to possible noises. In addition, although sampling schemes based on loss values require only lightweight calculations compared to those of e.g. gradient-weighted sampling, they basically ignore higher-order directional information in the training dynamics of the latter. In this regard, to additionally exploit the higher-order information (like gradient-weighted sampling) while maintaining the lightweight calculations (of loss-weighted sampling), we propose to use the sampling scheme Eq. (3.1) based on Prop. 3.1.

**Proposition 3.1.** *For any $i \in [n]$ and $t \in \mathbb{N}$, define the sampling probability as*

$$p_i(t) \propto w_i(t) = \beta_1 s_i(t-1) + (1-\beta_1)\ell_i(\boldsymbol{\theta}(t)),$$
$$s_i(t) = \beta_2 s_i(t-1) + (1-\beta_2)\ell_i(\boldsymbol{\theta}(t)) \tag{3.1}$$

*with $s_i(0) = 1/n$, and $\beta_1, \beta_2 \in [0,1]$ as two hyper-parameters (commonly $\beta_1 \leq \beta_2$). Then for any $\beta_2 \neq 1$, we have*

$$w_i(t) = (1-\beta_2)\sum_{k=1}^{t}\beta_2^{t-k}\ell_i(\boldsymbol{\theta}(k)) + (\beta_2 - \beta_1)\sum_{k=1}^{t-1}\beta_2^{t-1-k}\underbrace{(\ell_i(\boldsymbol{\theta}(k+1)) - \ell_i(\boldsymbol{\theta}(k)))}_{\textit{losses' dynamical differences}} + O(\beta_2^t). \tag{3.2}$$

*Furthermore, consider the SGD optimizer with the sampling scheme Eq. (3.1), i.e. $\boldsymbol{\theta}(t+1) = \boldsymbol{\theta}(t) - \eta_t \sum_{j\in\mathcal{B}_t} p_j(t)\nabla_{\boldsymbol{\theta}}\ell_j(\boldsymbol{\theta}(t))$. Then, for $t \gg 1$, we have*

$$w_i(t) \approx (1-\beta_2)\sum_{k=1}^{t}\beta_2^{t-k}\ell_i(\boldsymbol{\theta}(k))$$
$$- (\beta_2 - \beta_1)\sum_{k=1}^{t-1}\beta_2^{t-1-k}\eta_k\left\langle\nabla_{\boldsymbol{\theta}}\ell_i(\boldsymbol{\theta}(k)), \sum_{j\in\mathcal{B}_k}p_j(k)\nabla_{\boldsymbol{\theta}}\ell_j(\boldsymbol{\theta}(k))\right\rangle, \tag{3.3}$$

*where $c_i(k) := \left\langle\nabla_{\boldsymbol{\theta}}\ell_i(\boldsymbol{\theta}(k)), \sum_{j\in\mathcal{B}_k}p_j(k)\nabla_{\boldsymbol{\theta}}\ell_j(\boldsymbol{\theta}(k))\right\rangle$ denotes the inner product between the $i$-th sample's gradient and full gradient at the $k$-th iteration.*

The proof is deferred to Appx. B.2. Intuitively, $c_i(k)$ represents the individual-to-whole gradient "alignment" along training trajectories. As shown in Eq. (3.3), despite that there are only calculations regarding values of losses in the sampling scheme Eq. (3.1), Eq. (3.1) *implicitly* leverages additional *correlations between gradients* to determine sample weights: When the individual gradient positively correlates with the whole gradient (i.e. the better-learned sample with in step learning directions), we have $c_i(k) > 0$, and its sample weight is decreased as the second term of $w_i(t)$ is negative; conversely, when the individual gradient negatively correlates with the whole gradient (i.e. the worse-learned sample without in step learning directions), we have $c_i(k) < 0$, and its sample weight is increased as the second term of $w_i(t)$ is positive.

We discuss more implications of Prop. 3.1 as follows:

- The sampling scheme Eq. (3.1) reduces to Eq. (2.3) when setting $\beta_1 = \beta_2 = 0$,[2] hence it is an extension by augmenting the information of losses' dynamical differences.
- Prop. 3.1 suggests that one can incorporate additional dynamical variations of losses into the calculation of sampling weights through Eq. (3.1), *without explicitly storing historical losses and calculating differences* (as in Eq. (3.2)), making Eq. (3.1) an efficient sampling scheme by saving both memory and computation compared to Eq. (3.2).

---

[2]Also, it is obvious that Eq. (3.1) reduces to the standard batched sampling when setting $\beta_1 = \beta_2 = 1$.

- Based on Eq. (3.2), the strengths of losses and their dynamical differences can be flexibly balanced via the hyper-parameters $(\beta_1, \beta_2)$. By setting $(\beta_1, \beta_2) \to (0^+, 1^-)$, we are able to exploit the long-term historical information during training (via $\beta_2$), while simultaneously responding to current losses (via $\beta_1$) and thus can get the best of both world.[3]

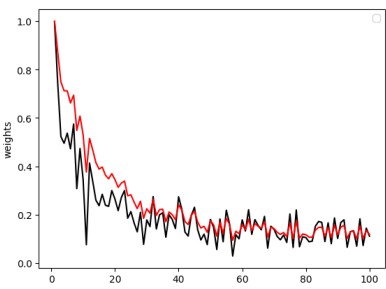

Figure 1: The output weights of different sampling schemes, where the black curve denotes Eq. (2.3), while the red curve represents Eq. (3.1) $((\beta_1, \beta_2) = (0.5, 0.9))$. Here, we draw the black curve as a decayed function with random perturbations, to mimic typical behaviors of loss curves in general machine learning. It is shown that the sampling scheme Eq. (2.3) is sensitive w.r.t. oscillations. However, when losses oscillate, the sampling scheme Eq. (3.1) reacts moderately by not only reserving some portion of dynamical details of losses (high frequency information), but also remaining necessary robustness by capturing the overall trend (low frequency information), with the flexibility to trade off in between by tuning $(\beta_1, \beta_2)$. See theoretical analysis in Sec. 3.2.

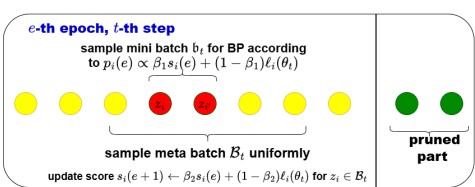

Figure 2: An illustration of ES(WP). At the beginning of the $e$-th epoch, we optionally randomly prune the whole dataset ("pruning"), *according to the probability proportional to the weights $\{w_i(e)\}_{i=1}^n$ defined in Eq. (3.1)*. At the $t$-th step, we first sample a meta-batch $\mathcal{B}_t$ uniformly without replacement from the remaining dataset; then we sample a mini-batch $\mathfrak{b}_t$ from $\mathcal{B}_t$ for BP, according to the sampling probability $p_i(\cdot)$ defined in Eq. (3.1). Note that the scores/weights of samples are updated using the *latest* model parameters. At the first/last few epochs, we optionally use the "annealing" strategy (Qin et al. (2024)), i.e. the standard batched sampling without data selection. See the algorithm details in Appx. C.

**More intuitions.** We further explain why augmented (loss) differences should work intuitively. Let $\beta_2 > \beta_1$. Given any data sample $z_i$, if its total loss variations accumulated up to $t$ are positive (say, $\ell_i(\cdot)$ always increases), the augmented "difference" term in Eq. (3.2) is positive and hence its sampling weight is increased, which is reasonable since the model continually underfits $z_i$ and should then value $z_i$ more. Conversely, if its loss variations accumulated up to $t$ are negative (say, $\ell_i(\cdot)$ always decreases), the augmented "difference" term in Eq. (3.2) is negative and hence its sampling weight is decreased, which is also reasonable since the model continually fits $z_i$ well and should then value $z_i$ less. That is, the augmented "difference" term in Eq. (3.2) plays a role of "damping". More quantitative justifications can be derived via the frequency analysis (see Sec. 3.2).

We establish the following estimate on the convergence rate of SGD weighted by the sampling scheme Eq. (3.1), and its proof is deferred in Appx. B.3.

**Theorem 3.2.** *Assume that $\ell_i(\cdot)$ is convex and $L_i$-smooth (i.e. $\|\nabla_{\boldsymbol{\theta}}\ell_i(\boldsymbol{\theta}_1) - \nabla_{\boldsymbol{\theta}}\ell_i(\boldsymbol{\theta}_2)\| \leq L_i\|\boldsymbol{\theta}_1 - \boldsymbol{\theta}_2\|$) for each $i \in [n]$, and there exists $\boldsymbol{\theta}^*$ such that $\hat{L}_n(\boldsymbol{\theta}^*) = 0$. Then, for the SGD optimizer with the sampling scheme Eq. (3.1), i.e. $\boldsymbol{\theta}(t+1) = \boldsymbol{\theta}(t) - \eta_t \sum_{j \in \mathcal{B}_t} p_j(t)\nabla_{\boldsymbol{\theta}}\ell_j(\boldsymbol{\theta}(t))$, with the constant learning rate $\eta_t \equiv \eta = 1/(2L)$ ($L := \max_{i \in [n]} L_i$), we have*

$$\hat{L}_n\left(\frac{1}{T}\sum_{t=0}^{T-1}\boldsymbol{\theta}(t)\right) - \hat{L}_n(\boldsymbol{\theta}^*) \leq \frac{2L\mathbb{E}\|\boldsymbol{\theta}(0) - \boldsymbol{\theta}^*\|_2^2}{T} - \frac{1}{T}\sum_{t=0}^{T-1}R(t), \quad (3.4)$$

*where $R(t) := \mathbb{E}\sum_{j \in \mathcal{B}_t}\left(p_j(t) - \frac{1}{B}\right)[\ell_j(\boldsymbol{\theta}(t)) - \ell_j(\boldsymbol{\theta}^*)]$ denotes the remainder. Furthermore, there exists $(\beta_1, \beta_2)$ such that $R(t) \geq 0$ for any $t \in \mathbb{N}$, leading to the more-than sub-linear convergence.*

---

[3]In fact, by Eq. (3.1), it is obvious that smaller $\beta_1$ and larger $\beta_2$ give larger coefficients of the current loss $\ell_i(\boldsymbol{\theta}(t))$ and historical score $s_i(t-1)$, respectively, hence we are focusing on the importance of both current losses and historical weights by setting $(\beta_1, \beta_2) \to (0^+, 1^-)$.

**Remark 1.** *Discussions on assumptions:*

- *The convexity and L-smoothness conditions are widely-adopted and also standard in proving the convergence of (S)GD in optimization literature (e.g. Garrigos & Gower (2023)). Under the non-convex setting, typically the* convergence to only stationary points *can be guaranteed for general smooth functions (see e.g. Khaled & Richtárik (2023)).*

- *For the condition $\hat{L}_n(\boldsymbol{\theta}^*) = 0$, it simply means that the optimal training loss can be zero. There are empirical evidences to support this assumption even for neural networks (see e.g. Figure 1 (a) in Zhang et al. (2017)).*

**ES(WP) framework.** We refer the scheme Eq. (3.1) as Evolved Sampling (ES), which conducts data selection on batch level. To further incorporate set level selection, we extend ES to prune data at each epoch, leading to **E**volved **S**ampling **W**ith **P**runing (ESWP) framework as illustrated in Fig. 2. Note that we optionally adopt annealing techniques to enhance performance. For the essential differences between ES(WP) and previous dynamic sampling methods, one can refer to the taxonomy outlined in Tab. 1. As a plug-and-play framework, ES(WP) can be integrated into any optimizers applied to different tasks, while some recently developed sampling methods (Wang et al. (2024a; 2025)) only work for SGD. In practice, the simple and elegant design of the sampling scheme Eq. (3.1) turns out to be surprisingly effective, as shown in extensive experiments.

**Remark 2.** *Here, we allow the randomness to keep samples with lower weights in training, which reduces the biases (and possibly inactive samples) compared to directly discarding them.*

## 3.2 THEORETICAL BENEFITS OF EVOLVED SAMPLING VIA FOURIER ANALYSIS

In this section, we provide mathematical justifications for the sampling scheme Eq. (3.1) by characterizing its frequency properties. To achieve this, we first view $\ell(t)(:= \ell(\boldsymbol{\theta}(t)))$, $s(t)$, $w(t)$ defined in the sampling scheme Eq. (3.1) all as signals in time. For ease of notation, we omit the sample index $i$ here. For a signal in time $f(t)$ (with appropriate regularities), we consider the Laplace transform $\mathcal{L}\{f\}(\omega) = \int_0^\infty e^{-\omega t} f(t)\mathrm{d}t$, $\omega \in \mathbb{C}$. Then, according to the Fourier analysis, $|\mathcal{L}\{f\}(\mathrm{i}\omega_0)|$ represents the magnitude of $f$'s $\omega_0$-frequency for $\omega_0 > 0$ ($\mathrm{i}^2 = -1$). We have the following result.

**Theorem 3.3.** *Consider a continuous-time idealization of the sampling scheme Eq. (3.1):*

$$w(t) = s(t) + \frac{\beta_2 - \beta_1}{1 - \beta_2} s'(t), \qquad s'(t) = (1 - \beta_2)(\ell(t) - s(t)), \tag{3.5}$$

*with $s(0) = 1/n$, and $\beta_1, \beta_2 \in (0, 1)$ as two hyper-parameters. Then we have*

$$\mathcal{L}\{w\}(\omega) = \frac{(\beta_2 - \beta_1)\omega + (1 - \beta_2)}{\omega + (1 - \beta_2)} \mathcal{L}\{\ell\}(\omega) + O(1/n), \tag{3.6}$$

*implying that the transfer function $H(\omega) = \frac{(\beta_2 - \beta_1)\omega + (1 - \beta_2)}{\omega + (1 - \beta_2)}$ satisfies*

$$|H(\mathrm{i}\omega_0)| \le 1, \qquad \lim_{\omega_0 \to +\infty} |H(\mathrm{i}\omega_0)| = |\beta_2 - \beta_1|. \tag{3.7}$$

The proof is provided in Appx. B.4. Based on Thm. 3.3, we conclude that (i) for all frequencies in the loss signal $\ell(t) = \ell(\boldsymbol{\theta}(t))$, the weight signal $w(t)$ calculated by the sampling scheme Eq. (3.1) does not enlarge them, hence is more stable in the frequency domain given oscillations in loss signals; (ii) for high frequencies in the loss signal $\ell(t) = \ell(\boldsymbol{\theta}(t))$, the weight signal $w(t)$ calculated by the sampling scheme Eq. (3.1) reserves a $|\beta_2 - \beta_1|$-portion, which can be tuned via betas (frequency tuning). This suggests that the sampling scheme Eq. (3.1) can not only stably capture the overall trend (low frequency), but also flexibly tune the portion of details (high frequency) in loss signals. See illustrations in Fig. 2 and Fig. 8.

## 3.3 UNVEILING THE ACCELERATION EFFECTS VIA COMPLEXITY ANALYSIS

**Computation efficiency.** The primary source of savings comes from the substantial reduction in the effective batch size during BP, compared with standard sampling (no data selection). Although ES(WP) introduces an extra forward pass (FP) on the selected mini-batch (can be *omitted* if selection is performed only at the set level, e.g., ESWP), the overhead is modest since FP requires much fewer FLOPs than BP. Consequently, the reduction in BP dominates the overall time complexity, leading to a significant acceleration effect, as observed empirically in Sec. 4.1.

**Memory efficiency.**    From Eq. (3.1), the only additional memory cost of ES(WP) is to store the current score/weight value of each sample, which is negligible compared to the memory cost high-dimensional data itself. Moreover, because ES(WP) reduces the effective sample size in BP, it further decreases memory usage (also verified numerically in Sec. 4.1).

**More significant benefits under resource constraints.**    The advantages of ES(WP) become even more significant in low-resource scenarios where GPU memory is limited and gradient accumulation is required, a typical scenario in fine-tuning large models (e.g., LLMs). In this setting, multiple BP passes must be executed before completing a single model update. Suppose the micro-batch size on each GPU is $b_{\text{micro}}$. Then under standard sampling, the number of BP per update step is $\lceil B/b_{\text{micro}} \rceil$. In contrast, ES(WP) requires only $\lceil b/b_{\text{micro}} \rceil$ BP passes. When $b_{\text{micro}} \leq b$, the time spent on BP under standard sampling can be up to $B/b$ times greater than with ES(WP), underscoring the stronger acceleration of our method in memory-constrained training.

### 3.4 HYPER-PARAMETERS TUNING

The primary hyper-parameters are betas in the sampling scheme Eq. (3.1), which are designed to balance dynamical losses and their differences during training. In experiments, we take the default values of $(\beta_1, \beta_2)$, which are obtained by a fine-grained grid search in small-scale simulations (Sec. 4.3). These defaults are consistently validated to be (locally) optimal in small-scale experiments, and their superior effectiveness remains in large-scale tasks (Sec. 4.1, (ii) & (iii), Sec. 4.2).[4] The other hyper-parameters, including mini-batch sizes, the pruning ratio and annealing epochs, are all responsible for trade-offs between the learning performance and training speed. All of them are user-defined, similar to previous data selection methods such as Qin et al. (2024); Thao Nguyen et al. (2023); Raju et al. (2021). We also perform comprehensive ablations in Sec. 4.3.

## 4 EXPERIMENTS

In this section, we provide numerical simulations on the proposed method ES(WP) to demonstrate its superiority in terms of effectiveness, efficiency, robustness and flexibility.[5]

### 4.1 EFFECTIVENESS AND EFFICIENCY

We compare the proposed methods ES/ESWP, with a group of former dynamic sampling approaches, including the standard batched sampling (Baseline), purely random pruning (Random), Ordered SGD (Order; Kawaguchi & Lu (2020)), Loss (Katharopoulos & Fleuret (2017), i.e., the sampling scheme Eq. (2.3)), InfoBatch (Qin et al. (2024)), KAKURENBO (KA; Thao Nguyen et al. (2023)), UCB (Raju et al. (2021)). For fair comparisons, all these sampling methods are loss-based, hence *much more light-weighted than gradient-based ones*, and *do not require to (pre-)train or exploit additional models*. See Appx. A, Paragraph "Dynamic sampling" for detailed discussions. For all sampling methods, the hyper-parameters used in data pre-processing and optimization follow standard configurations and are maintained the same (see more details in Appx. D). All reported results are evaluated on the average of 3-4 independent random trials.

**Configurations.**    For ES/ESWP, the default hyper-parameters are as follows: In Eq. (3.1), $(\beta_1, \beta_2) = (0.2, 0.9)$ for ES, $(\beta_1, \beta_2) = (0.2, 0.8)$ for ESWP; for both ES and ESWP, the ratio of mini-batch size ($b$) over meta-batch size ($B$) is $b/B = 25\%$; if applicable, the annealing ratio is $5\%$, i.e., no data selection is performed at the first/last $5\%$ epochs; for ESWP (and Random), the pruning ratio is $20\%$. For the two batch level data selection methods (Order, Loss), we apply the same mini/meta-batch size as ES(WP). For InfoBatch, KA and UCB (set level data selection methods), we use the default hyper-parameters in original papers (see more details in Appx. D.7).

---

[4]Notably, here we follow *a common routine* of hyper-parameters tuning, which is also adopted in e.g. Qin et al. (2024); Wang et al. (2024b); Thao Nguyen et al. (2023), to reuse default hyper-parameters (obtained by grid search in small-scale simulations) in large-scale experiments, without further tuning. This also indicates that the joint effect of betas is robust, and the gain of ES(WP) is not from simply introducing/tuning more hyper-parameters, but essentially from the augmented losses' differences.

[5]We will release the code after acceptance.

**Results.** We report the test classification accuracy and overall wall-clock time for the evaluation of both effectiveness and efficiency. The results are as follows.

(i) For small-scale tasks, we train ResNet models on CIFAR datasets, and summarize the performance of different sampling methods in Tab. 2. It is shown that the batch level data selection methods (Loss, Order, ES) typically exhibit limited accelerations on these small-scale tasks, since these methods often require additional forward propagation overheads that are not negligible compared to BPs. Nevertheless, ES is the only algorithm that achieves lossless accelerations across all sampling methods. Notably, ESWP saves the most computation time while maintaining the best performance (also comparable to Baseline) among set level data selection methods (UCB, KA, InfoBatch).

Table 2: The test accuracy (%) and saved time of training ResNet models on CIFAR datasets.

|  | CIFAR-10 (R-18) | | CIFAR-100 (R-18) | | CIFAR-100 (R-50) | |
| --- | --- | --- | --- | --- | --- | --- |
| Baseline | 95.4 | | 78.8 | | 81.1 | |
| Random | $95.4_{\downarrow0.1}$ | 18% | $78.4_{\downarrow0.4}$ | 20% | $80.8_{\downarrow0.3}$ | 19% |
| UCB (Raju et al. (2021)) | $95.2_{\downarrow0.2}$ | 18% | $77.6_{\downarrow1.2}$ | 18% | $80.5_{\downarrow0.6}$ | 24% |
| KA (Thao Nguyen et al. (2023)) | $95.3_{\downarrow0.1}$ | 21% | $78.1_{\downarrow0.7}$ | 21% | $80.2_{\downarrow0.9}$ | 24% |
| InfoBatch (Qin et al. (2024)) | $95.3_{\downarrow0.1}$ | 21% | $78.4_{\downarrow0.4}$ | 24% | $80.4_{\downarrow0.7}$ | 28% |
| Loss (Katharopoulos & Fleuret (2017)) | $95.3_{\downarrow0.1}$ | 11% | $78.4_{\downarrow0.4}$ | 10% | $80.5_{\downarrow0.6}$ | 12% |
| Order (Kawaguchi & Lu (2020)) | $95.4_{\uparrow0.0}$ | 11% | $78.5_{\downarrow0.3}$ | 10% | $80.9_{\downarrow0.2}$ | 12% |
| ES | $95.4_{\uparrow0.0}$ | 10% | $78.8_{\uparrow0.0}$ | 10% | $81.1_{\uparrow0.0}$ | 11% |
| ESWP | $95.3_{\downarrow0.1}$ | 24% | $78.6_{\downarrow0.2}$ | 24% | $80.6_{\downarrow0.5}$ | 31% |

Table 3: The test accuracy and saved time of fully fine-tuning ViT-Large on ImageNet-1K.

|  | Time ↓ | Acc. (%) |
| --- | --- | --- |
| Baseline | – | 84.4 |
| Random | 24.5% | $84.5_{\uparrow0.1}$ |
| UCB | 23.6% | $84.2_{\downarrow0.2}$ |
| KA | 25.3% | $84.3_{\downarrow0.1}$ |
| InfoBatch | 23.5% | $84.7_{\uparrow0.3}$ |
| Loss | 36.4% | $84.3_{\downarrow0.2}$ |
| Order | 38.2% | $84.2_{\downarrow0.2}$ |
| ES | 26.0% | $84.7_{\uparrow0.3}$ |
| ESWP | **40.7%** | $\mathbf{85.0}_{\uparrow0.6}$ |

*Selected samples by ES(WP).* In Appx. D.2, we provide a visualization of selected samples by ES(WP). Following Mindermann et al. (2022), we also plot the test accuracy versus the number of samples used for back propagations (BPs) for Baseline and ES/ESWP in Fig. 10. It is clear that ES/ESWP can significantly reduce the BP calculation costs and thus improve the learning efficiency.

(ii) For large-scale tasks, we fully fine-tune ViT-Large on ImageNet-1K, and summarize the performance in Tab. 3. Under this setting, ES consistently achieves the best performance among batch level data selection methods and the second-to-highest accuracy across all sampling methods. Notably, ESWP attains the highest accuracy and the most significant wall-clock time reduction, suggesting that ESWP inherits the advantages of *both* set and batch level data selection methods. In addition, it is observed that the training speed-up of batch level methods gets far more significant given these large-scale tasks, conversely surpassing the set level methods compared to (i). This is due to the dominance of the saved computation in BPs. Furthermore, many sampling methods achieve higher accuracies than Baseline, implying the huge potential of data selection in large-scale deep learning. We also numerically evaluate the corresponding overall memory usage of ES (49.7GB) and ESWP (49.1GB), which are also reduced compared to Baseline (52.4GB), verifying the efficiency of ES(WP) in terms of memory loads besides computation costs for large-scale tasks.

(iii) For (large-scale) distributed learning tasks, we pre-train ViT-Large using MAE (He et al. (2022)), and then fine-tune on ImageNet-1K without data selection. We plot the re-construction loss curves in Fig. 3 and report final accuracy after fine-tuning in Tab. 4. It is shown that ESWP achieves lossless acceleration over Baseline, and consistently outperforms the previous SOTA method InfoBatch.

(iv) For NLP tasks, we fully fine-tune ALBERT-Base-v2 on GLUE, and summarize the results in Tab. 5. Across most datasets and on average, ES/ESWP outperforms all the other sampling methods, and shows improved performance over Baseline with significant reduction of computation time.

## 4.2 Low-Resource Settings: More Acceleration in LLM Fine-tuning

In this section, we investigate the low-resource setting by fine-tuning Qwen2.5-Math-1.5B (Yang et al. (2024)) on a single NVIDIA A100 (40GB). We sample 30K instances from NuminaMath CoT (LI et al. (2024)), and conduct SFT with a maximum token length 1024 and thus $b_{\text{micro}} = 8$. We set $B = 32, b = 8$ and the pruning ratio as 0.2 for ESWP. The averaged evaluation results on MATH500 (Hendrycks et al. (2021)), AIME24, and OlympiadBench (He et al. (2024)) are shown in Fig. 4, where we evaluate the model after 1K, 2K, and 4K training steps. Under this low-resource setting, the time cost of BPs is significant due to gradient accumulations, whereas ESWP can reduce this cost by selecting a much smaller effective mini-batch, thereby achieving learning accelerations. This

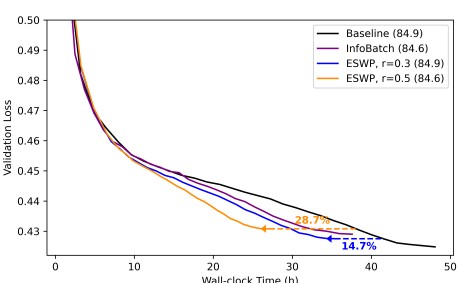

Figure 3: Re-construction losses of MAE-based pre-training of ViT-Large on ImageNet-1K. The number in the bracket in the legend is the validation accuracy (%) after fine-tuning, and $r$ stands for the pruning ratio.

Table 4: Pre-training time and fine-tuning accuracy.

|  | Time (h) | Time ↓ | Acc. (%) |
|---|---|---|---|
| Baseline | 48.1 | – | 84.9 |
| InfoBatch | 37.6 | 21.8% | $84.6_{\downarrow 0.3}$ |
| ESWP ($r = 0.3$) | 35.1 | 27.0% | $\mathbf{84.9}_{\uparrow 0.0}$ |
| ESWP ($r = 0.5$) | **27.1** | **44.7%** | $84.6_{\downarrow 0.3}$ |

Table 5: The validation metric (%) and saved time of fully fine-tuning ALBERT-Base-v2 on GLUE.

|  | CoLA | SST2 | QNLI | QQP | MNLI-m | MRPC | RTE | STSB | Avg. | Time↓ |
|---|---|---|---|---|---|---|---|---|---|---|
| Baseline | 56.7 | 92.2 | 91.1 | **90.3** | **84.7** | 88.5 | 74.0 | 89.6 | 83.4 | - |
| InfoBatch | 57.9 | 92.1 | 91.2 | **90.3** | 84.5 | 89.2 | 73.8 | **89.7** | $83.6_{\uparrow 0.2}$ | 28.3% |
| Loss | 55.1 | 92.3 | 91.4 | 90.2 | 84.4 | 88.6 | 69.6 | 89.5 | $82.6_{\downarrow 0.8}$ | 20.8% |
| Order | 55.4 | 92.6 | 91.3 | 90.1 | 80.9 | 84.6 | 63.2 | 89.4 | $80.9_{\downarrow 2.5}$ | 20.8% |
| ES | **58.4** | 92.4 | 91.4 | 90.2 | 84.5 | 88.7 | **75.8** | 89.6 | $\mathbf{83.9}_{\uparrow 0.5}$ | 20.2% |
| ESWP | 57.5 | **93.1** | **91.7** | 90.0 | **84.7** | 89.8 | 72.8 | 89.4 | $83.6_{\uparrow 0.2}$ | **33.1%** |

highlights the superiority of ESWP in computation-constrained and memory-limited environments, where ESWP shows accelerations with improved performance compared to Baseline. More details are provided in Appx. D.6.

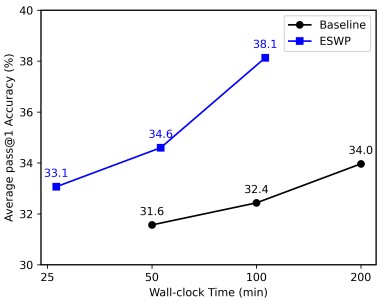

Figure 4: The evaluation results of Qwen SFT averaged on MATH500, AIME24, and OlympiadBench under the low-resource setting.

## 4.3 Ablation Studies

**Loss differences, annealing and pruning.** We numerically test the effectiveness of the most important component adopted in ES(WP), i.e. the augmented dynamical differences of losses. For completeness, we also test the effect of the annealing technique and pruning strategy. Here, we perform ablations on combinations of "Loss" (the sampling scheme Eq. (2.3)), $\beta_1 = \beta_2 = 0$), "NonDif" (corresponding to $\beta_1 = \beta_2$, see Eq. (3.2)), "Dif" (Eq. (3.1)), corresponding to general betas $\beta_1 \neq \beta_2$) and "A" (Annealing). From Tab. 6, it is observed that: (i) Annealing is an effective technique to boost performance; (ii) Although sampling only involving historical losses ("NonDif") can contribute to the improvements, the additional incorporation of dynamical loss differences consistently provides more substantial benefits to the learning process (see consistent non-trivial improvements for various datasets and models in the last two rows of Tab. 6). In Tab. 7, we further ablate for the pruning strategies: Eq. (3.1) used in ESWP versus naive random data pruning. It is shown that both the performance and efficiency of purely random pruning are consistently and substantially worse than ESWP.

Table 6: Ablations on the effect of augmented dynamical differences of losses and annealing.

|  | ResNet-50 CIFAR-100 | ALBERT-Base CoLA |
|---|---|---|
| Loss | 80.5 | 55.1 |
| Loss + A | 80.8 | 55.8 |
| Loss + NonDif | 80.5 | 57.6 |
| Loss + Dif | **81.1** | 57.5 |
| Loss + A + NonDif | 80.4 | 57.6 |
| ES = Loss + A + Dif | **81.1** | **58.4** |

Table 7: Ablations on pruning strategies. Here Random denotes purely random data pruning.

|  | CoLA ALBERT-Base | SST-2 ALBERT-Base |
|---|---|---|
| Baseline | 55.0, − | 91.9, − |
| Random | $53.9_{\downarrow 1.1}, 18\%$ | $91.7_{\downarrow 0.2}, 20\%$ |
| ES | $\mathbf{56.2}_{\uparrow 1.2}, 16\%$ | $92.0_{\uparrow 0.1}, 15\%$ |
| ESWP | $54.7_{\downarrow 0.3}, 24\%$ | $\mathbf{92.3}_{\uparrow 0.4}, 24\%$ |

**Trade-offs between performance and speed.** We emphasize that batch sizes $(b, B)$, the pruning ratio and annealing epochs in ES(WP) are all user-defined, and flexible to trade off between learning performance and training costs. We evaluate different values of $b/B$ when fine-tuning ViT-Large on ImageNet-1K, and varied pruning ratios when training R-18 on Cifar-100. The results are illustrated in Fig. 5. It is shown that ES robustly achieves lossless acceleration when $b/B \geq 1/16$; when the data selection is too aggressive ($b/B \leq 1/32$), the performance degrades as expected (Fig. 5, left), due to the increase of variances in stochastic gradients. Also, there is a clear trade-off between the performance and speed shown in Fig. 5 (right), where setting the pruning ratio around $20\% \sim 30\%$ seems efficient. We further evaluate different annealing ratios ("ar"; i.e., annealing epochs over total epochs) when training R-18 on CIFAR-100 (see Tab. 8), showcasing its robustness.

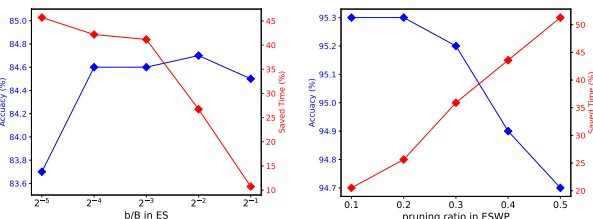

Table 8: Ablations on the annealing (default in bold).

| ar | 0.0 | **0.05** | 0.075 | 0.1 |
|---|---|---|---|---|
| Acc. (%) | 78.60 | **78.79** | 78.32 | 78.20 |

Figure 5: Trade-offs between learning accuracy and wall-clock time.

**Choices of $(\beta_1, \beta_2)$.** To investigate the impact of newly introduced hyper-parameters (betas) in ES, we test different choices of $(\beta_1, \beta_2)$ when training ResNet-18 on CIFAR datasets and ALBERT-Base model on the CoLA dataset. The results shown in Fig. 6 roughly verify the "optimality" of defaults ($(\beta_1, \beta_2) = (0.2, 0.9)$). In addition, we test denser betas around the defaults when training ResNet-18 on the CIFAR-100 (see Fig. 7), further verifying the (local) optimality of defaults.

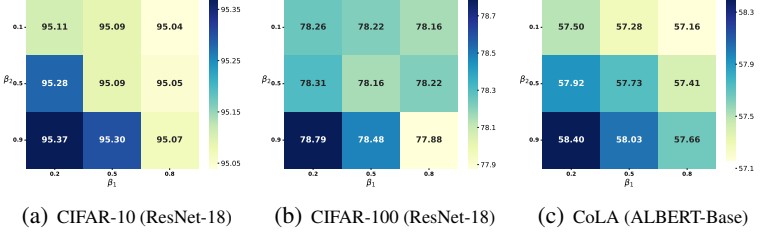

(a) CIFAR-10 (ResNet-18)    (b) CIFAR-100 (ResNet-18)    (c) CoLA (ALBERT-Base)

Figure 7: Local optimality of default betas.

Figure 6: The effect of $(\beta_1, \beta_2)$.

## 5 CONCLUSION

In this work, we propose a simple yet effective framework, Evolved Sampling, which can be applied to general machine learning tasks to improve the data efficiency in a dynamic manner. By further adopting dynamical differences and flexibly tuning frequencies of historical losses to determine samples' importance for data selection, Evolved Sampling can achieve lossless training with significant accelerations. Studies in the future may include three aspects: (i) More rigorous mathematical analysis on the effect of data selection (Kolossov et al. (2024)); (ii) More specific applications, such as data selection on domain mixtures (Chen et al. (2023); Xie et al. (2023a)); (iii) More efficient and scalable implementations, such as data parallelism (You et al. (2017; 2020)). These directions are certainly worthy of explorations in the future.

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

## A  RELATED WORK

**Static sampling**    Methods to sampling statically can be based on geometry, uncertainty, error, meta optimization, dataset distillation, etc. With numerous studies on theoretical guarantees (Har-Peled & Mazumdar (2004); Huang et al. (2023); Bachem et al. (2015)), the coreset selection is designed to approximate original datasets with smaller (weighted) subsets, typically achieved by clustering in representation spaces (Xia et al. (2023); Abbas et al. (2023); Sorscher et al. (2022)). Uncertainty-based methods use probability metrics such as the confidence, entropy (Coleman et al. (2020)) and distances to decision boundaries (Ducoffe & Precioso (2018); Margatina et al. (2021); Dasgupta et al. (2019); Liu et al. (2021)). Sampling methods based on errors assume that training samples with

more contributions to errors are more important. Errors are evaluated with merics such as forgetting events (Toneva et al. (2019)), GRAND & EL2N score (Paul et al. (2021)), and sensitivity (Langberg & Schulman (2010); Munteanu et al. (2018)). The meta optimization methods apply a bilevel framework to learn the re-weighting. In general, existing studies such as RETRIEVE (Killamsetty et al. (2021c)), GLISTER (Killamsetty et al. (2021b)), MOLERE (Jain et al. (2024)), CAFE (Wang et al. (2022)) and so on, consider the data selection as the outer objective (over selection weights), and the optimization of model parameters on selected subsets as the inner objective. Dataset distillation aims to synthesize an informative but smaller subset from the original (large) dataset. Although there are multiple implementations to reduce the overall loads, such as distributed kernel computation (Nguyen et al. (2021)), category decoupling (Wang et al. (2022)), random modeling (Zhao & Bilen (2023)) and so on, the dataset distillation still requires to optimize over both the model and data, and is hence expensive. A recent work Sachdeva et al. (2024) also leverages the zero-shot reasoning capability of instruction-tuned large language models (LLMs) to directly assess the quality of data examples. As is discussed before, these static sampling methods require extra training, leading to considerable costs in both computation and memory.

**Dynamic sampling**    Methods to sampling dynamically typically leverage metrics based on losses and gradients along the training process. Loss-adaptive sampling re-weights data points during the training according to current losses (Jiang et al. (2019); Loshchilov & Hutter (2016); Schaul et al. (2016); Kawaguchi & Lu (2020); Qin et al. (2024); Thao Nguyen et al. (2023); Kumar et al. (2023); Balaban et al. (2023); Katharopoulos & Fleuret (2017); Shrivastava et al. (2016); Das et al. (2024)) and historical losses (Attendu & Corbeil (2023); Raju et al. (2021); Oren et al. (2019); Sagawa et al. (2020)). To name a few, Ordered SGD (Kawaguchi & Lu (2020)) selects top-$q$ samples in terms of the loss ranking per training step. InfoBatch (Qin et al. (2024)) randomly prunes a portion of less informative samples with losses below the average and then re-scales the gradients. KAKURENBO (Thao Nguyen et al. (2023)) combines current losses with the prediction accuracy and confidence to design a sampling framework with moving-back. Kumar et al. (2023) and Balaban et al. (2023) assign weights as functions of current losses based on the robust optimization theory. Attendu & Corbeil (2023) and Raju et al. (2021) use the exponential moving average over past losses for sampling. There are also studies adopting additional reference models, including Mindermann et al. (2022); Deng et al. (2023); Xie et al. (2023a) and so on. These methods either use the information of losses inadequately, or require to train or exploit extra architectures. Gradient-based sampling methods involve (i) gradient matching, such as CRAIG (Mirzasoleiman et al. (2020)) and GRAD-MATCH (Killamsetty et al. (2021a)), which approximate the "full" gradients computed on original datasets via the gradients computed on subsets; (ii) gradient adaption, where the sampling probability is basically determined by current scales of gradients (Hanchi et al. (2022); Katharopoulos & Fleuret (2018)). Obviously, gradient-based sampling methods lead to much more computation and memory overheads than loss-based methods. A recent work Wang et al. (2024b) uses a intricate layer-wise sampling scheme with complex variance control, which develops former literature Zhao & Zhang (2015); Alain et al. (2015); Needell et al. (2014) applying importance sampling methods to accelerate the convergence by reducing variances. A very recent work Gu et al. (2025) also leverages the optimal control theory (i.e. Pontryagin's maximum principle, PMP) to formulate and decide sampling weights, where both the gradient and Hessian are computed and all historical checkpoints are stored. Obviously, these methods usually suffer from significant computation and memory loads, since extra complexities of at least model dimensions are introduced at every training step. Although there are other gradient-based data selection methods (e.g. Wang et al. (2024a): local approximation-based selection; Wang et al. (2025): counterfactual-based selection) developing computation reduction techniques such as the ghost inner-product (of gradients) and generalized Gauss-Newton approximation (of Hessians), these methods are not directly extendable to other popular optimizers like Adam.

***Set* level versus *batch* level**    Dynamic sampling methods can be divided into two categories based on the level where data selection is performed: (i) *set* level selection, to prune the whole dataset at the beginning of each epoch (Qin et al. (2024); Raju et al. (2021); Thao Nguyen et al. (2023); Attendu & Corbeil (2023)); (ii) *batch* level selection, to sample subsets from the original batches for back propagations (Kawaguchi & Lu (2020); Katharopoulos & Fleuret (2017; 2018); Mindermann et al. (2022)). These two types of methods, facilitating training accelerations from different perspectives, are not mutually exclusive. However, to the best of our knowledge, we are not aware of any algorithms combining both of them.

# B PROOFS AND SUPPLEMENTAL THEORY

## B.1 PROOF OF PROP. 2.1

**Proposition B.1** (A full version of Prop. 2.1). *Consider the continuous-time idealization of the gradient decent, i.e. the standard gradient flow training dynamics (no data selection)*

$$\frac{\mathrm{d}}{\mathrm{d}t}\hat{\boldsymbol{\theta}}_n(t) = -\nabla_{\boldsymbol{\theta}}\hat{L}_n(\hat{\boldsymbol{\theta}}_n(t)) = -\frac{1}{n}\sum_{i=1}^n \nabla_{\boldsymbol{\theta}}\ell_i(\hat{\boldsymbol{\theta}}_n(t)), \quad \hat{\boldsymbol{\theta}}_n(0) = \boldsymbol{\theta}_0, \tag{B.1}$$

*and its loss-weighted variant*

$$\frac{\mathrm{d}}{\mathrm{d}s}\hat{\boldsymbol{\theta}}_n^{lw}(s) = -\sum_{i=1}^n \frac{\ell_i(\hat{\boldsymbol{\theta}}_n^{lw}(s))}{\sum_{j=1}^n \ell_j(\hat{\boldsymbol{\theta}}_n^{lw}(s))}\nabla_{\boldsymbol{\theta}}\ell_i(\hat{\boldsymbol{\theta}}_n^{lw}(s)), \quad \hat{\boldsymbol{\theta}}_n^{lw}(0) = \boldsymbol{\theta}_0. \tag{B.2}$$

*Assume that there exists $\boldsymbol{\theta}^* \in \Theta$ such that $\hat{L}_n(\boldsymbol{\theta}^*) = 0$,[6] and $\ell_i(\cdot)$ is convex for each $i \in [n]$. Then, we have the more-than sub-linear convergence rate of Eq. (B.2), i.e., there exists $s_0 \in [0, s]$ such that*

$$\hat{L}_n(\hat{\boldsymbol{\theta}}_n^{lw}(s_0)) - \hat{L}_n(\boldsymbol{\theta}^*) \leq \frac{1}{2s}\|\boldsymbol{\theta}_0 - \boldsymbol{\theta}^*\|_2^2 - \frac{1}{s}\int_0^s \Delta(s')\mathrm{d}s', \quad s > 0, \tag{B.3}$$

*where $\Delta(\cdot)$ is a positive-valued function on $[0, \infty)$. Moreover, for any $s, t \geq 0$ such that $\hat{L}_n(\hat{\boldsymbol{\theta}}_n(t)) = \hat{L}_n(\hat{\boldsymbol{\theta}}_n^{lw}(s)) \triangleq l \geq 0$,[7] we have*

$$\frac{\mathrm{d}}{\mathrm{d}s}\|\hat{\boldsymbol{\theta}}_n^{lw}(s) - \boldsymbol{\theta}^*\|_2^2 \leq -2\left(l + \Delta(s)\right), \tag{B.4}$$

$$\frac{\mathrm{d}}{\mathrm{d}t}\|\hat{\boldsymbol{\theta}}_n(t) - \boldsymbol{\theta}^*\|_2^2 \leq -2l. \tag{B.5}$$

*Proof.* For any $\boldsymbol{\theta} \in \Theta$, by convexity we have

$$\begin{aligned}
\frac{\mathrm{d}}{\mathrm{d}t}\|\hat{\boldsymbol{\theta}}_n(t) - \boldsymbol{\theta}\|_2^2 &= 2\left\langle \hat{\boldsymbol{\theta}}_n(t) - \boldsymbol{\theta}, \frac{\mathrm{d}}{\mathrm{d}t}\hat{\boldsymbol{\theta}}_n(t) \right\rangle \\
&= \frac{2}{n}\sum_{i=1}^n \left\langle \boldsymbol{\theta} - \hat{\boldsymbol{\theta}}_n(t), \nabla_{\boldsymbol{\theta}}\ell_i(\hat{\boldsymbol{\theta}}_n(t)) \right\rangle \\
&\leq \frac{2}{n}\sum_{i=1}^n \left(\ell_i(\boldsymbol{\theta}) - \ell_i(\hat{\boldsymbol{\theta}}_n(t))\right),
\end{aligned} \tag{B.6}$$

and

$$\begin{aligned}
\frac{\mathrm{d}}{\mathrm{d}s}\|\hat{\boldsymbol{\theta}}_n^{\mathrm{lw}}(s) - \boldsymbol{\theta}\|_2^2 &= 2\left\langle \hat{\boldsymbol{\theta}}_n^{\mathrm{lw}}(s) - \boldsymbol{\theta}, \frac{\mathrm{d}}{\mathrm{d}s}\hat{\boldsymbol{\theta}}_n^{\mathrm{lw}}(s) \right\rangle \\
&= 2\sum_{i=1}^n \frac{\ell_i(\hat{\boldsymbol{\theta}}_n^{\mathrm{lw}}(s))}{\sum_{j=1}^n \ell_j(\hat{\boldsymbol{\theta}}_n^{\mathrm{lw}}(s))}\left\langle \boldsymbol{\theta} - \hat{\boldsymbol{\theta}}_n^{\mathrm{lw}}(s), \nabla_{\boldsymbol{\theta}}\ell_i(\hat{\boldsymbol{\theta}}_n^{\mathrm{lw}}(s)) \right\rangle \\
&\leq 2\sum_{i=1}^n \frac{\ell_i(\hat{\boldsymbol{\theta}}_n^{\mathrm{lw}}(s))}{\sum_{j=1}^n \ell_j(\hat{\boldsymbol{\theta}}_n^{\mathrm{lw}}(s))}\left(\ell_i(\boldsymbol{\theta}) - \ell_i(\hat{\boldsymbol{\theta}}_n^{\mathrm{lw}}(s))\right).
\end{aligned} \tag{B.7}$$

Note that

$$\begin{aligned}
&\sum_{i=1}^n \left[\frac{\ell_i(\hat{\boldsymbol{\theta}}_n^{\mathrm{lw}}(s))}{\sum_{j=1}^n \ell_j(\hat{\boldsymbol{\theta}}_n^{\mathrm{lw}}(s))}\left(\ell_i(\boldsymbol{\theta}) - \ell_i(\hat{\boldsymbol{\theta}}_n^{\mathrm{lw}}(s))\right) - \frac{1}{n}\left(\ell_i(\boldsymbol{\theta}) - \ell_i(\hat{\boldsymbol{\theta}}_n(t))\right)\right] \\
&= \sum_{i=1}^n \left(\frac{\ell_i(\hat{\boldsymbol{\theta}}_n^{\mathrm{lw}}(s))}{\sum_{j=1}^n \ell_j(\hat{\boldsymbol{\theta}}_n^{\mathrm{lw}}(s))} - \frac{1}{n}\right)\left(\ell_i(\boldsymbol{\theta}) - \ell_i(\hat{\boldsymbol{\theta}}_n^{\mathrm{lw}}(s))\right) + \frac{1}{n}\sum_{i=1}^n \left(\ell_i(\hat{\boldsymbol{\theta}}_n(t)) - \ell_i(\hat{\boldsymbol{\theta}}_n^{\mathrm{lw}}(s))\right)
\end{aligned}$$

---

[6] One can find empirical evidences of this assumption (the optimal training loss can be zero) in e.g. Zhang et al. (2017) (Figure 1 (a)).

[7] For example, at the initialization, $\hat{L}_n(\hat{\boldsymbol{\theta}}_n(0)) = \hat{L}_n(\boldsymbol{\theta}_0) = \hat{L}_n(\hat{\boldsymbol{\theta}}_n^{\mathrm{lw}}(0))$.

$$= - \sum_{i=1}^{n} \underbrace{\left( \frac{\ell_i(\hat{\boldsymbol{\theta}}_n^{\mathrm{lw}}(s))}{\sum_{j=1}^{n} \ell_j(\hat{\boldsymbol{\theta}}_n^{\mathrm{lw}}(s))} - \frac{1}{n} \right) \left( \ell_i(\hat{\boldsymbol{\theta}}_n^{\mathrm{lw}}(s)) - \ell_i(\boldsymbol{\theta}) \right)}_{T_1} + \underbrace{\left( \hat{L}_n(\hat{\boldsymbol{\theta}}_n(t)) - \hat{L}_n(\hat{\boldsymbol{\theta}}_n^{\mathrm{lw}}(s)) \right)}_{T_2}, \quad \text{(B.8)}$$

we analyze $T_1, T_2$ separately.

(i) $T_1$: Note that if $\frac{\ell_i(\hat{\boldsymbol{\theta}}_n^{\mathrm{lw}}(s))}{\sum_{j=1}^{n} \ell_j(\hat{\boldsymbol{\theta}}_n^{\mathrm{lw}}(s))} \leq \frac{1}{n}$ for any $i \in [n]$, we get $\frac{\ell_i(\hat{\boldsymbol{\theta}}_n^{\mathrm{lw}}(s))}{\sum_{j=1}^{n} \ell_j(\hat{\boldsymbol{\theta}}_n^{\mathrm{lw}}(s))} = \frac{1}{n}$ for any $i \in [n]$, which holds in the zero probability and implies the triviality. Let $I^+ := \left\{ i \in [n] : \frac{\ell_i(\hat{\boldsymbol{\theta}}_n^{\mathrm{lw}}(s))}{\sum_{j=1}^{n} \ell_j(\hat{\boldsymbol{\theta}}_n^{\mathrm{lw}}(s))} > \frac{1}{n} \right\} \neq \varnothing$, and $i_{\min}^+ := \arg\min_{i \in I^+} \ell_i(\hat{\boldsymbol{\theta}}_n^{\mathrm{lw}}(s))$, and similarly $I^- := \left\{ i \in [n] : \frac{\ell_i(\hat{\boldsymbol{\theta}}_n^{\mathrm{lw}}(s))}{\sum_{j=1}^{n} \ell_j(\hat{\boldsymbol{\theta}}_n^{\mathrm{lw}}(s))} \leq \frac{1}{n} \right\} \neq \varnothing$, and $i_{\max}^- := \arg\max_{i \in I^-} \ell_i(\hat{\boldsymbol{\theta}}_n^{\mathrm{lw}}(s))$. Obviously, $\ell_{i_{\min}^+}(\hat{\boldsymbol{\theta}}_n^{\mathrm{lw}}(s)) > \frac{1}{n} \sum_{j=1}^{n} \ell_j(\hat{\boldsymbol{\theta}}_n^{\mathrm{lw}}(s)) \geq \ell_{i_{\max}^-}(\hat{\boldsymbol{\theta}}_n^{\mathrm{lw}}(s))$, hence $\delta(s) := \ell_{i_{\min}^+}(\hat{\boldsymbol{\theta}}_n^{\mathrm{lw}}(s)) - \ell_{i_{\max}^-}(\hat{\boldsymbol{\theta}}_n^{\mathrm{lw}}(s)) > 0$ for any $s \geq 0$. Notice that $\hat{L}_n(\boldsymbol{\theta}^*) = 0 \Leftrightarrow \ell_i(\boldsymbol{\theta}^*) = 0, \forall i \in [n]$, we have

$$T_1\big|_{\boldsymbol{\theta}=\boldsymbol{\theta}^*} = \sum_{i \in I^+} \left( \frac{\ell_i(\hat{\boldsymbol{\theta}}_n^{\mathrm{lw}}(s))}{\sum_{j=1}^{n} \ell_j(\hat{\boldsymbol{\theta}}_n^{\mathrm{lw}}(s))} - \frac{1}{n} \right) \left( \ell_i(\hat{\boldsymbol{\theta}}_n^{\mathrm{lw}}(s)) - \ell_i(\boldsymbol{\theta}^*) \right)$$

$$+ \sum_{i \in I^-} \left( \frac{\ell_i(\hat{\boldsymbol{\theta}}_n^{\mathrm{lw}}(s))}{\sum_{j=1}^{n} \ell_j(\hat{\boldsymbol{\theta}}_n^{\mathrm{lw}}(s))} - \frac{1}{n} \right) \left( \ell_i(\hat{\boldsymbol{\theta}}_n^{\mathrm{lw}}(s)) - \ell_i(\boldsymbol{\theta}^*) \right)$$

$$= \sum_{i \in I^+} \left( \frac{\ell_i(\hat{\boldsymbol{\theta}}_n^{\mathrm{lw}}(s))}{\sum_{j=1}^{n} \ell_j(\hat{\boldsymbol{\theta}}_n^{\mathrm{lw}}(s))} - \frac{1}{n} \right) \ell_i(\hat{\boldsymbol{\theta}}_n^{\mathrm{lw}}(s)) + \sum_{i \in I^-} \left( \frac{\ell_i(\hat{\boldsymbol{\theta}}_n^{\mathrm{lw}}(s))}{\sum_{j=1}^{n} \ell_j(\hat{\boldsymbol{\theta}}_n^{\mathrm{lw}}(s))} - \frac{1}{n} \right) \ell_i(\hat{\boldsymbol{\theta}}_n^{\mathrm{lw}}(s))$$

$$\geq \sum_{i \in I^+} \left( \frac{\ell_i(\hat{\boldsymbol{\theta}}_n^{\mathrm{lw}}(s))}{\sum_{j=1}^{n} \ell_j(\hat{\boldsymbol{\theta}}_n^{\mathrm{lw}}(s))} - \frac{1}{n} \right) \ell_{i_{\min}^+}(\hat{\boldsymbol{\theta}}_n^{\mathrm{lw}}(s)) + \sum_{i \in I^-} \left( \frac{\ell_i(\hat{\boldsymbol{\theta}}_n^{\mathrm{lw}}(s))}{\sum_{j=1}^{n} \ell_j(\hat{\boldsymbol{\theta}}_n^{\mathrm{lw}}(s))} - \frac{1}{n} \right) \ell_{i_{\max}^-}(\hat{\boldsymbol{\theta}}_n^{\mathrm{lw}}(s))$$

$$= \sum_{i \in I^+} \left( \frac{\ell_i(\hat{\boldsymbol{\theta}}_n^{\mathrm{lw}}(s))}{\sum_{j=1}^{n} \ell_j(\hat{\boldsymbol{\theta}}_n^{\mathrm{lw}}(s))} - \frac{1}{n} \right) \left( \ell_{i_{\max}^-}(\hat{\boldsymbol{\theta}}_n^{\mathrm{lw}}(s)) + \delta(s) \right) + \sum_{i \in I^-} \left( \frac{\ell_i(\hat{\boldsymbol{\theta}}_n^{\mathrm{lw}}(s))}{\sum_{j=1}^{n} \ell_j(\hat{\boldsymbol{\theta}}_n^{\mathrm{lw}}(s))} - \frac{1}{n} \right) \ell_{i_{\max}^-}(\hat{\boldsymbol{\theta}}_n^{\mathrm{lw}}(s))$$

$$= \ell_{i_{\max}^-}(\hat{\boldsymbol{\theta}}_n^{\mathrm{lw}}(s)) \sum_{i=1}^{n} \left( \frac{\ell_i(\hat{\boldsymbol{\theta}}_n^{\mathrm{lw}}(s))}{\sum_{j=1}^{n} \ell_j(\hat{\boldsymbol{\theta}}_n^{\mathrm{lw}}(s))} - \frac{1}{n} \right) + \delta(s) \sum_{i \in I^+} \left( \frac{\ell_i(\hat{\boldsymbol{\theta}}_n^{\mathrm{lw}}(s))}{\sum_{j=1}^{n} \ell_j(\hat{\boldsymbol{\theta}}_n^{\mathrm{lw}}(s))} - \frac{1}{n} \right)$$

$$= \ell_{i_{\max}^-}(\hat{\boldsymbol{\theta}}_n^{\mathrm{lw}}(s))(1-1) + \Delta(s) = \Delta(s), \quad \text{(B.9)}$$

where $\Delta(s) := \delta(s) \sum_{i \in I^+} \left( \frac{\ell_i(\hat{\boldsymbol{\theta}}_n^{\mathrm{lw}}(s))}{\sum_{j=1}^{n} \ell_j(\hat{\boldsymbol{\theta}}_n^{\mathrm{lw}}(s))} - \frac{1}{n} \right) > 0$ for any $s \geq 0$. By continuity, $T_1\big|_{\boldsymbol{\theta}} \geq \Delta(s)/2 > 0$ also holds for any $\boldsymbol{\theta} \approx \boldsymbol{\theta}^*$.

(ii) $T_2$: It measures the difference between losses under the standard and loss-weighted gradient flow. Combining Eq. (B.7), Eq. (B.8) with Eq. (B.9) yields that

$$\frac{\mathrm{d}}{\mathrm{d}s} \|\hat{\boldsymbol{\theta}}_n^{\mathrm{lw}}(s) - \boldsymbol{\theta}^*\|_2^2 \leq 2 \left[ \frac{1}{n} \sum_{i=1}^{n} \left( \ell_i(\boldsymbol{\theta}^*) - \ell_i(\hat{\boldsymbol{\theta}}_n(t)) \right) - T_1\big|_{\boldsymbol{\theta}=\boldsymbol{\theta}^*} + T_2 \right]$$

$$\leq 2 \left[ \left( \hat{L}_n(\boldsymbol{\theta}^*) - \hat{L}_n(\hat{\boldsymbol{\theta}}_n(t)) \right) - \Delta(s) + \left( \hat{L}_n(\hat{\boldsymbol{\theta}}_n(t)) - \hat{L}_n(\hat{\boldsymbol{\theta}}_n^{\mathrm{lw}}(s)) \right) \right]$$

$$= 2 \left[ \left( \hat{L}_n(\boldsymbol{\theta}^*) - \hat{L}_n(\hat{\boldsymbol{\theta}}_n^{\mathrm{lw}}(s)) \right) - \Delta(s) \right], \quad \text{(B.10)}$$

which gives

$$\hat{L}_n(\hat{\boldsymbol{\theta}}_n^{\mathrm{lw}}(s)) - \hat{L}_n(\boldsymbol{\theta}^*) \leq -\frac{1}{2} \frac{\mathrm{d}}{\mathrm{d}s} \|\hat{\boldsymbol{\theta}}_n^{\mathrm{lw}}(s) - \boldsymbol{\theta}^*\|_2^2 - \Delta(s) \quad \text{(B.11)}$$

$$\Rightarrow \int_{s_1}^{s_2} \hat{L}_n(\hat{\boldsymbol{\theta}}_n^{\mathrm{lw}}(s)) \mathrm{d}s - (s_2 - s_1) \cdot \hat{L}_n(\boldsymbol{\theta}^*) \leq -\frac{1}{2} \left( \|\hat{\boldsymbol{\theta}}_n^{\mathrm{lw}}(s_2) - \boldsymbol{\theta}^*\|_2^2 - \|\hat{\boldsymbol{\theta}}_n^{\mathrm{lw}}(s_1) - \boldsymbol{\theta}^*\|_2^2 \right) - \int_{s_1}^{s_2} \Delta(s) \mathrm{d}s$$

$$\leq \frac{1}{2}\|\hat{\boldsymbol{\theta}}_n^{\text{lw}}(s_1) - \boldsymbol{\theta}^*\|_2^2 - \int_{s_1}^{s_2} \Delta(s)\mathrm{d}s \tag{B.12}$$

for any $s_2 > s_1 \geq 0$. That is

$$\frac{1}{s_2 - s_1}\int_{s_1}^{s_2} \hat{L}_n(\hat{\boldsymbol{\theta}}_n^{\text{lw}}(s))\mathrm{d}s - \hat{L}_n(\boldsymbol{\theta}^*) \leq \frac{1}{2(s_2 - s_1)}\|\hat{\boldsymbol{\theta}}_n^{\text{lw}}(s_1) - \boldsymbol{\theta}^*\|_2^2 - \frac{1}{s_2 - s_1}\int_{s_1}^{s_2} \Delta(s)\mathrm{d}s,$$

or for any $s > 0$,

$$\frac{1}{s}\int_0^s \hat{L}_n(\hat{\boldsymbol{\theta}}_n^{\text{lw}}(s'))\mathrm{d}s' - \hat{L}_n(\boldsymbol{\theta}^*) \leq \frac{1}{2s}\|\boldsymbol{\theta}_0 - \boldsymbol{\theta}^*\|_2^2 - \frac{1}{s}\int_0^s \Delta(s')\mathrm{d}s'$$

$$< \frac{1}{2s}\|\boldsymbol{\theta}_0 - \boldsymbol{\theta}^*\|_2^2, \tag{B.13}$$

which proves Eq. (B.3) by the mean value theorem of integrals. Recall that Eq. (B.6) can be rewritten as

$$\frac{\mathrm{d}}{\mathrm{d}t}\|\hat{\boldsymbol{\theta}}_n(t) - \boldsymbol{\theta}^*\|_2^2 \leq 2\left(\hat{L}_n(\boldsymbol{\theta}^*) - \hat{L}_n(\hat{\boldsymbol{\theta}}_n(t))\right). \tag{B.14}$$

Compared with Eq. (B.10), for any $s, t \geq 0$ such that $\hat{L}_n(\hat{\boldsymbol{\theta}}_n(t)) = \hat{L}_n(\hat{\boldsymbol{\theta}}_n^{\text{lw}}(s))$, we have Eq. (B.10)'s RHS < Eq. (B.14)'s RHS $= -2\hat{L}_n(\hat{\boldsymbol{\theta}}_n(t)) \leq 0$, which implies a sharper convergence bound of the loss-weighted gradient flow (at the same loss level with the standard gradient flow). The proof is completed. $\square$

Prop. B.1 suggests that, under certain regularity conditions, the loss-weighted gradient flow converges more than sub-linearly to the global minimum, while the standard gradient flow (i.e the continuous-time idealization of vanilla GD) only has the sub-linear convergence. In addition, at the same loss level, the convergence bound of loss-weighted gradient flow is sharper than that of standard gradient flow. This theoretical characterization, together with practical simulations (e.g., Table 1, 3 and Figure 3, 4 in Kawaguchi & Lu (2020)), fundamentally gives chances to potential learning accelerations by leveraging the loss information in the gradient-based training dynamics.

### B.2 PROOF OF PROP. 3.1

*Proof.* Define $\boldsymbol{w}(t) := [w_i(t)]_{i \in [n]}$, $\boldsymbol{s}(t) := [s_i(t)]_{i \in [n]}$, and $\boldsymbol{l}(t) := [\ell_i(\boldsymbol{\theta}(t))]_{i \in [n]}$ for any $t \in \mathbb{N}$. The sampling scheme Eq. (3.1) can be rewritten as

$$\begin{aligned} \boldsymbol{w}(t) &= \beta_1 \boldsymbol{s}(t-1) + (1-\beta_1)\boldsymbol{l}(t), \\ \boldsymbol{s}(t) &= \beta_2 \boldsymbol{s}(t-1) + (1-\beta_2)\boldsymbol{l}(t), \quad \boldsymbol{s}(0) = \mathbf{1}/n. \end{aligned} \tag{B.15}$$

In Eq. (B.15), let the first equation minus the second, we get

$$\boldsymbol{w}(t) - \boldsymbol{s}(t) = (\beta_2 - \beta_1)(\boldsymbol{l}(t) - \boldsymbol{s}(t-1)). \tag{B.16}$$

The second equation gives

$$\boldsymbol{s}(t) - \boldsymbol{s}(t-1) = (1-\beta_2)(\boldsymbol{l}(t) - \boldsymbol{s}(t-1)). \tag{B.17}$$

Combining Eq. (B.16) with Eq. (B.17), we have

$$\boldsymbol{w}(t) = \boldsymbol{s}(t) + \frac{\beta_2 - \beta_1}{1 - \beta_2}(\boldsymbol{s}(t) - \boldsymbol{s}(t-1)), \tag{B.18}$$

which proves the first equality.

Expanding the second equation, by induction we get

$$\boldsymbol{s}(t) = \beta_2^t \boldsymbol{s}(0) + (1-\beta_2)\sum_{k=1}^t \beta_2^{t-k}\boldsymbol{l}(k), \tag{B.19}$$

hence

$$\boldsymbol{s}(t) - \boldsymbol{s}(t-1) = \beta_2^{t-1}(\beta_2 - 1)\boldsymbol{s}(0) + (1-\beta_2)\left[\sum_{k=1}^t \beta_2^{t-k}\boldsymbol{l}(k) - \sum_{k=1}^{t-1}\beta_2^{t-1-k}\boldsymbol{l}(k)\right]$$

$$= -(1-\beta_2)\beta_2^{t-1}\boldsymbol{s}(0) + (1-\beta_2)\left[\beta_2^{t-1}\boldsymbol{l}(1) + \sum_{k=2}^{t}\beta_2^{t-k}\boldsymbol{l}(k) - \sum_{k=1}^{t-1}\beta_2^{t-1-k}\boldsymbol{l}(k)\right]$$

$$= -(1-\beta_2)\beta_2^{t-1}\boldsymbol{s}(0) + (1-\beta_2)\left[\beta_2^{t-1}\boldsymbol{l}(1) + \sum_{k=1}^{t-1}\beta_2^{t-1-k}(\boldsymbol{l}(k+1) - \boldsymbol{l}(k))\right]$$

$$\approx (1-\beta_2)\sum_{k=1}^{t-1}\beta_2^{t-1-k}(\boldsymbol{l}(k+1) - \boldsymbol{l}(k)) \tag{B.20}$$

for relatively large $t$, and the approximation error is exponentially small (due to $\lim_{t\to+\infty}\beta_2^t = 0$ for any $\beta_2 \in (0,1)$). Combining Eq. (B.18), Eq. (B.19) and Eq. (B.20) yields Eq. (3.2).

Given the stochastic gradient descent (SGD) training dynamics, the model parameters are updated by

$$\boldsymbol{\theta}(t+1) = \boldsymbol{\theta}(t) - \eta_t \sum_{j\in\mathcal{B}_t} p_j(t)\nabla_{\boldsymbol{\theta}}\ell_j(\boldsymbol{\theta}(t)), \tag{B.21}$$

where $\{\eta_t\}_{t\in\mathbb{N}}$ denotes learning rates, and $\mathcal{B}_t \subset [n]$ denotes the subset of $\{1, 2, \cdots, n\}$, with the (batch) size $|\mathcal{B}_t| = B$. Then, by Taylor expansion, we have

$$\ell_i(\boldsymbol{\theta}(k+1)) - \ell_i(\boldsymbol{\theta}(k)) = \langle\nabla_{\boldsymbol{\theta}}\ell_i(\boldsymbol{\theta}(k)), \boldsymbol{\theta}(k+1) - \boldsymbol{\theta}(k)\rangle + O(\|\boldsymbol{\theta}(k+1) - \boldsymbol{\theta}(k)\|_2^2) \tag{B.22}$$

$$= -\eta_k c_i(k) + O(\|\boldsymbol{\theta}(k+1) - \boldsymbol{\theta}(k)\|_2^2), \tag{B.23}$$

where $c_i(k) := \langle\nabla_{\boldsymbol{\theta}}\ell_i(\boldsymbol{\theta}(k)), \sum_{j\in\mathcal{B}_k} p_j(k)\nabla_{\boldsymbol{\theta}}\ell_j(\boldsymbol{\theta}(k))\rangle$ denotes the inner product between the $i$-th sample's gradient and full gradient at the $k$-th iteration, representing the individual-to-whole gradient "alignment" along training trajectories. This yields

$$(\beta_2 - \beta_1)\sum_{k=1}^{t-1}\beta_2^{t-1-k}(\ell_i(\boldsymbol{\theta}(k+1)) - \ell_i(\boldsymbol{\theta}(k)))$$

$$= -(\beta_2 - \beta_1)\sum_{k=1}^{t-1}\beta_2^{t-1-k}[\eta_k c_i(k) + O(\|\boldsymbol{\theta}(k+1) - \boldsymbol{\theta}(k)\|_2^2)]$$

$$\overset{(i)}{\approx} -(\beta_2 - \beta_1)\sum_{k=1}^{t-1}\beta_2^{t-1-k}\eta_k c_i(k) - (\beta_2 - \beta_1)\sum_{k=t-O(1)}\beta_2^{t-1-k}\cdot O(\|\boldsymbol{\theta}(k+1) - \boldsymbol{\theta}(k)\|_2^2)]$$

$$\overset{(ii)}{\approx} -(\beta_2 - \beta_1)\sum_{k=1}^{t-1}\beta_2^{t-1-k}\eta_k c_i(k), \tag{B.24}$$

where (i) is due to the fact that $\beta_2^s$ ($\beta_2 \in (0,1)$) is exponentially small for relatively large $s > 0$ and $t \gg 1$, and (ii) is a consequence of convergence. The proof is completed. $\square$

## B.3 Proof of Thm. 3.2

We begin by proving some lemmas. The first two lemmas and their proofs are standard, and can be found in e.g. Garrigos & Gower (2023).

**Lemma B.2.** *If $f : \mathbb{R}^d \mapsto \mathbb{R}$ is $L$-smooth, then for any $\boldsymbol{x}, \boldsymbol{y} \in \mathbb{R}^d$, we have*

$$f(\boldsymbol{y}) \le f(\boldsymbol{x}) + \langle\nabla f(\boldsymbol{x}), \boldsymbol{y} - \boldsymbol{x}\rangle + \frac{L}{2}\|\boldsymbol{y} - \boldsymbol{x}\|_2^2. \tag{B.25}$$

*Proof.* For any fixed $\boldsymbol{x}, \boldsymbol{y} \in \mathbb{R}^d$, let $\phi(t) := f(\boldsymbol{x} + t(\boldsymbol{y} - \boldsymbol{x}))$. Then we have

$$f(\boldsymbol{y}) - f(\boldsymbol{x}) = \phi(1) - \phi(0) = \int_0^1 \phi'(t)\mathrm{d}t = \int_0^1 \langle\nabla f(\boldsymbol{x} + t(\boldsymbol{y} - \boldsymbol{x})), \boldsymbol{y} - \boldsymbol{x}\rangle\,\mathrm{d}t$$

$$= \langle \nabla f(\boldsymbol{x}), \boldsymbol{y} - \boldsymbol{x} \rangle + \int_0^1 \langle \nabla f(\boldsymbol{x} + t(\boldsymbol{y} - \boldsymbol{x})) - \nabla f(\boldsymbol{x}), \boldsymbol{y} - \boldsymbol{x} \rangle \, \mathrm{d}t$$

$$\leq \langle \nabla f(\boldsymbol{x}), \boldsymbol{y} - \boldsymbol{x} \rangle + \int_0^1 \|\nabla f(\boldsymbol{x} + t(\boldsymbol{y} - \boldsymbol{x})) - \nabla f(\boldsymbol{x})\|_2 \|\boldsymbol{y} - \boldsymbol{x}\|_2 \mathrm{d}t$$

$$\leq \langle \nabla f(\boldsymbol{x}), \boldsymbol{y} - \boldsymbol{x} \rangle + \int_0^1 Lt \|\boldsymbol{y} - \boldsymbol{x}\|_2^2 \mathrm{d}t$$

$$= \langle \nabla f(\boldsymbol{x}), \boldsymbol{y} - \boldsymbol{x} \rangle + \frac{L}{2} \|\boldsymbol{y} - \boldsymbol{x}\|_2^2, \tag{B.26}$$

where Cauchy–Schwarz inequality and the $L$-smoothness property are successively applied. The proof is completed. □

**Lemma B.3.** *If $f : \mathbb{R}^d \mapsto \mathbb{R}$ is convex and $L$-smooth, then for any $\boldsymbol{x}, \boldsymbol{y} \in \mathbb{R}^d$, we have*

$$\frac{1}{2L} \|\nabla f(\boldsymbol{y}) - \nabla f(\boldsymbol{x})\|_2^2 \leq f(\boldsymbol{y}) - f(\boldsymbol{x}) - \langle \nabla f(\boldsymbol{x}), \boldsymbol{y} - \boldsymbol{x} \rangle. \tag{B.27}$$

*Proof.* Fix any $\boldsymbol{x}, \boldsymbol{y} \in \mathbb{R}^d$. By convexity and $L$-smoothness, for any $\boldsymbol{z} \in \mathbb{R}^d$, we have

$$f(\boldsymbol{x}) - f(\boldsymbol{y}) = f(\boldsymbol{x}) - f(\boldsymbol{z}) + f(\boldsymbol{z}) - f(\boldsymbol{y})$$

$$\leq \langle \nabla f(\boldsymbol{x}), \boldsymbol{x} - \boldsymbol{z} \rangle + \langle \nabla f(\boldsymbol{y}), \boldsymbol{z} - \boldsymbol{y} \rangle + \frac{L}{2} \|\boldsymbol{z} - \boldsymbol{y}\|_2^2. \quad \text{(Lem. B.2)} \tag{B.28}$$

Take $\boldsymbol{z} = \boldsymbol{y} - (\nabla f(\boldsymbol{y}) - \nabla f(\boldsymbol{x}))/L$ to minimize the right hand side, we get

$$f(\boldsymbol{x}) - f(\boldsymbol{y}) \leq \langle \nabla f(\boldsymbol{x}), \boldsymbol{x} - \boldsymbol{y} \rangle - \frac{1}{2L} \|\nabla f(\boldsymbol{y}) - \nabla f(\boldsymbol{x})\|_2^2, \tag{B.29}$$

which completes the proof. □

We also need the following norm estimate on the product between matrices and probability-like vectors.

**Lemma B.4.** *For any matrix $\boldsymbol{G} := [\boldsymbol{g}_1, \cdots, \boldsymbol{g}_n] \in \mathbb{R}^{m \times n}$, and any vector $\boldsymbol{p} := [p_1, \cdots, p_n]^\top \in \mathbb{R}^n$ satisfying $\sum_{i=1}^n p_i \leq 1$ with $p_i \geq 0$ for all $i \in [n]$, we have*

$$\|\boldsymbol{G}\boldsymbol{p}\|_2^2 = \left\| \sum_{i=1}^n p_i \boldsymbol{g}_i \right\|_2^2 \leq \sum_{i=1}^n p_i \|\boldsymbol{g}_i\|_2^2. \tag{B.30}$$

*Proof.* It is straightforward to verify that

$$\|\boldsymbol{G}\boldsymbol{p}\|_2^2 = \left( \sum_{i=1}^n p_i \boldsymbol{g}_i \right)^\top \sum_{j=1}^n p_j \boldsymbol{g}_j = \sum_{i=1}^n \sum_{j=1}^n p_i p_j \boldsymbol{g}_i^\top \boldsymbol{g}_j$$

$$\leq \frac{1}{2} \sum_{i=1}^n \sum_{j=1}^n p_i p_j (\|\boldsymbol{g}_i\|_2^2 + \|\boldsymbol{g}_j\|_2^2) \leq \sum_{i=1}^n p_i \|\boldsymbol{g}_i\|_2^2, \tag{B.31}$$

which completes the proof. □

Now we are ready to prove the main theorem.

*Proof.* Given the SGD training dynamics $\boldsymbol{\theta}(t+1) = \boldsymbol{\theta}(t) - \eta_t \sum_{j \in \mathcal{B}_t} p_j(t) \nabla_{\boldsymbol{\theta}} \ell_j(\boldsymbol{\theta}(t))$, we have

$$\|\boldsymbol{\theta}(t+1) - \boldsymbol{\theta}^*\|_2^2 = \|(\boldsymbol{\theta}(t+1) - \boldsymbol{\theta}(t)) + (\boldsymbol{\theta}(t) - \boldsymbol{\theta}^*)\|_2^2$$

$$= \|\boldsymbol{\theta}(t+1) - \boldsymbol{\theta}(t)\|_2^2 + \|\boldsymbol{\theta}(t) - \boldsymbol{\theta}^*\|_2^2 + 2 \langle \boldsymbol{\theta}(t+1) - \boldsymbol{\theta}(t), \boldsymbol{\theta}(t) - \boldsymbol{\theta}^* \rangle$$

$$= \|\boldsymbol{\theta}(t) - \boldsymbol{\theta}^*\|_2^2 + \left\| \eta_t \sum_{j \in \mathcal{B}_t} p_j(t) \nabla_{\boldsymbol{\theta}} \ell_j(\boldsymbol{\theta}(t)) \right\|_2^2 - 2 \left\langle \eta_t \sum_{j \in \mathcal{B}_t} p_j(t) \nabla_{\boldsymbol{\theta}} \ell_j(\boldsymbol{\theta}(t)), \boldsymbol{\theta}(t) - \boldsymbol{\theta}^* \right\rangle$$

$$=: \|\boldsymbol{\theta}(t) - \boldsymbol{\theta}^*\|_2^2 + I_1 + I_2. \tag{B.32}$$

For $I_1$, by Lem. B.4 we have

$$I_1 = \eta_t^2 \left\| \sum_{j \in \mathcal{B}_t} p_j(t) \nabla_{\boldsymbol{\theta}} \ell_j(\boldsymbol{\theta}(t)) \right\|_2^2 \le \eta_t^2 \sum_{j \in \mathcal{B}_t} p_j(t) \|\nabla_{\boldsymbol{\theta}} \ell_j(\boldsymbol{\theta}(t))\|_2^2. \tag{B.33}$$

Due to the optimality of $\boldsymbol{\theta}^*$, we have $\nabla_{\boldsymbol{\theta}} \ell_i(\boldsymbol{\theta}^*) = 0$ for any $i \in [n]$. Recall that $\ell_i(\cdot)$ is $L_i$-smooth, we derive by Lem. B.3 that

$$\begin{aligned} \|\nabla_{\boldsymbol{\theta}} \ell_j(\boldsymbol{\theta}(t))\|_2^2 &= \|\nabla_{\boldsymbol{\theta}} \ell_j(\boldsymbol{\theta}(t)) - \nabla_{\boldsymbol{\theta}} \ell_j(\boldsymbol{\theta}^*)\|_2^2 \\ &\le 2L_j \left[ \ell_j(\boldsymbol{\theta}(t)) - \ell_j(\boldsymbol{\theta}^*) - \langle \nabla \ell_j(\boldsymbol{\theta}^*), \boldsymbol{\theta}(t) - \boldsymbol{\theta}^* \rangle \right] \\ &= 2L_j \left[ \ell_j(\boldsymbol{\theta}(t)) - \ell_j(\boldsymbol{\theta}^*) \right], \end{aligned} \tag{B.34}$$

which gives

$$I_1 \le 2\eta_t^2 \sum_{j \in \mathcal{B}_t} L_j p_j(t) \left[ \ell_j(\boldsymbol{\theta}(t)) - \ell_j(\boldsymbol{\theta}^*) \right]. \tag{B.35}$$

For $I_2$, by convexity we have

$$\begin{aligned} I_2 &= 2\eta_t \sum_{j \in \mathcal{B}_t} p_j(t) \langle \nabla_{\boldsymbol{\theta}} \ell_j(\boldsymbol{\theta}(t)), \boldsymbol{\theta}^* - \boldsymbol{\theta}(t) \rangle \\ &\le 2\eta_t \sum_{j \in \mathcal{B}_t} p_j(t)(\ell_j(\boldsymbol{\theta}^*) - \ell_j(\boldsymbol{\theta}(t))) \\ &= -2\eta_t \sum_{j \in \mathcal{B}_t} p_j(t) \left[ \ell_j(\boldsymbol{\theta}(t)) - \ell_j(\boldsymbol{\theta}^*) \right]. \end{aligned} \tag{B.36}$$

Therefore, we obtain

$$\|\boldsymbol{\theta}(t+1) - \boldsymbol{\theta}^*\|_2^2 \le \|\boldsymbol{\theta}(t) - \boldsymbol{\theta}^*\|_2^2 + 2\eta_t \sum_{j \in \mathcal{B}_t} (\eta_t L_j - 1) p_j(t) \left[ \ell_j(\boldsymbol{\theta}(t)) - \ell_j(\boldsymbol{\theta}^*) \right]. \tag{B.37}$$

Let $L := \max_{i \in [n]} L_i$, and set $\eta_t \le 1/(2L)$. Then, take the expectation (conditioned on $(\boldsymbol{\theta}(s))_{s \le t}$) over both sides of Eq. (B.37), we have

$$\begin{aligned} \mathbb{E} \|\boldsymbol{\theta}(t+1) - \boldsymbol{\theta}^*\|_2^2 &\le \|\boldsymbol{\theta}(t) - \boldsymbol{\theta}^*\|_2^2 + 2\eta_t \sum_{j \in \mathcal{B}_t} (\eta_t L_j - 1) p_j(t) \left[ \ell_j(\boldsymbol{\theta}(t)) - \ell_j(\boldsymbol{\theta}^*) \right] \\ &\le \|\boldsymbol{\theta}(t) - \boldsymbol{\theta}^*\|_2^2 - \eta_t \sum_{j \in \mathcal{B}_t} p_j(t) \left[ \ell_j(\boldsymbol{\theta}(t)) - \ell_j(\boldsymbol{\theta}^*) \right] \\ &= \|\boldsymbol{\theta}(t) - \boldsymbol{\theta}^*\|_2^2 - \eta_t \frac{1}{B} \sum_{j \in \mathcal{B}_t} \left[ \ell_j(\boldsymbol{\theta}(t)) - \ell_j(\boldsymbol{\theta}^*) \right] \\ &\quad - \eta_t \sum_{j \in \mathcal{B}_t} \left( p_j(t) - \frac{1}{B} \right) \left[ \ell_j(\boldsymbol{\theta}(t)) - \ell_j(\boldsymbol{\theta}^*) \right], \end{aligned} \tag{B.38}$$

and by law of total expectation,

$$\begin{aligned} \mathbb{E} \|\boldsymbol{\theta}(t+1) - \boldsymbol{\theta}^*\|_2^2 &\le \mathbb{E} \|\boldsymbol{\theta}(t) - \boldsymbol{\theta}^*\|_2^2 - \eta_t \mathbb{E} \frac{1}{B} \sum_{j \in \mathcal{B}_t} \left[ \ell_j(\boldsymbol{\theta}(t)) - \ell_j(\boldsymbol{\theta}^*) \right] \\ &\quad - \eta_t \mathbb{E} \sum_{j \in \mathcal{B}_t} \left( p_j(t) - \frac{1}{B} \right) \left[ \ell_j(\boldsymbol{\theta}(t)) - \ell_j(\boldsymbol{\theta}^*) \right] \\ &= \mathbb{E} \|\boldsymbol{\theta}(t) - \boldsymbol{\theta}^*\|_2^2 - \eta_t \left[ \hat{L}_n(\boldsymbol{\theta}(t)) - \hat{L}_n(\boldsymbol{\theta}^*) \right] \\ &\quad - \eta_t \mathbb{E} \sum_{j \in \mathcal{B}_t} \left( p_j(t) - \frac{1}{B} \right) \left[ \ell_j(\boldsymbol{\theta}(t)) - \ell_j(\boldsymbol{\theta}^*) \right]. \end{aligned} \tag{B.39}$$

Then by telescoping sum, we obtain

$$\mathbb{E} \left\| \boldsymbol{\theta}(T) - \boldsymbol{\theta}^* \right\|_2^2 \leq \mathbb{E} \left\| \boldsymbol{\theta}(0) - \boldsymbol{\theta}^* \right\|_2^2 - \sum_{t=0}^{T-1} \eta_t \left[ \hat{L}_n(\boldsymbol{\theta}(t)) - \hat{L}_n(\boldsymbol{\theta}^*) \right]$$

$$- \sum_{t=0}^{T-1} \eta_t \mathbb{E} \sum_{j \in \mathcal{B}_t} \left( p_j(t) - \frac{1}{B} \right) \left[ \ell_j(\boldsymbol{\theta}(t)) - \ell_j(\boldsymbol{\theta}^*) \right], \quad \text{(B.40)}$$

which yields

$$\sum_{t=0}^{T-1} \eta_t \left[ \hat{L}_n(\boldsymbol{\theta}(t)) - \hat{L}_n(\boldsymbol{\theta}^*) \right] \leq \mathbb{E} \left\| \boldsymbol{\theta}(0) - \boldsymbol{\theta}^* \right\|_2^2 - \sum_{t=0}^{T-1} \eta_t R(t), \quad \text{(B.41)}$$

where $R(t) := \mathbb{E} \sum_{j \in \mathcal{B}_t} \left( p_j(t) - \frac{1}{B} \right) \left[ \ell_j(\boldsymbol{\theta}(t)) - \ell_j(\boldsymbol{\theta}^*) \right]$ denotes the remainder. Therefore

$$\sum_{t=0}^{T-1} \frac{\eta_t}{\sum_{s=0}^{T-1} \eta_s} \left[ \hat{L}_n(\boldsymbol{\theta}(t)) - \hat{L}_n(\boldsymbol{\theta}^*) \right] \leq \frac{\mathbb{E} \left\| \boldsymbol{\theta}(0) - \boldsymbol{\theta}^* \right\|_2^2}{\sum_{s=0}^{T-1} \eta_s} - \sum_{t=0}^{T-1} \frac{\eta_t}{\sum_{s=0}^{T-1} \eta_s} R(t), \quad \text{(B.42)}$$

and by convexity

$$\hat{L}_n(\bar{\boldsymbol{\theta}}_T) - \hat{L}_n(\boldsymbol{\theta}^*) \leq \frac{\mathbb{E} \left\| \boldsymbol{\theta}(0) - \boldsymbol{\theta}^* \right\|_2^2}{\sum_{s=0}^{T-1} \eta_s} - \sum_{t=0}^{T-1} \frac{\eta_t}{\sum_{s=0}^{T-1} \eta_s} R(t), \quad \text{(B.43)}$$

with $\bar{\boldsymbol{\theta}}_T := \sum_{t=0}^{T-1} \frac{\eta_t}{\sum_{s=0}^{T-1} \eta_s} \boldsymbol{\theta}(t)$. For $\eta_t \equiv \eta = 1/(2L)$, Eq. (B.43) gives

$$\hat{L}_n \left( \frac{1}{T} \sum_{t=0}^{T-1} \boldsymbol{\theta}(t) \right) - \hat{L}_n(\boldsymbol{\theta}^*) \leq \frac{2L \mathbb{E} \left\| \boldsymbol{\theta}(0) - \boldsymbol{\theta}^* \right\|_2^2}{T} - \frac{1}{T} \sum_{t=0}^{T-1} R(t). \quad \text{(B.44)}$$

Next, we provide a *sufficient* condition to bound the remainder term $\frac{1}{T} \sum_{t=0}^{T-1} R(t)$ (from below). For instance, in the sampling scheme Eq. (3.1), there exists $(\beta_1, \beta_2)$ such that

$$(w_i(t) - w_j(t))(\ell_i(\boldsymbol{\theta}(t)) - \ell_j(\boldsymbol{\theta}(t))) \geq 0 \quad \text{(B.45)}$$

for any $i, j \in [n]$ and $t \in \mathbb{N}$ (e.g. when $\beta_1 \to 0^+$).[8] Therefore, for any $t \in \mathbb{N}$, we have

$$0 \leq \sum_{i=1}^{B} \sum_{j=1}^{B} (p_i(t) - p_j(t))(\ell_i(\boldsymbol{\theta}(t)) - \ell_j(\boldsymbol{\theta}(t)))$$

$$= \sum_{i=1}^{B} \sum_{j=1}^{B} (p_i(t)\ell_i(\boldsymbol{\theta}(t)) + p_j(t)\ell_j(\boldsymbol{\theta}(t)) - p_i(t)\ell_j(\boldsymbol{\theta}(t)) - p_j(t)\ell_i(\boldsymbol{\theta}(t)))$$

$$= B \sum_{i=1}^{B} p_i(t)\ell_i(\boldsymbol{\theta}(t)) + B \sum_{j=1}^{B} p_j(t)\ell_j(\boldsymbol{\theta}(t)) - \sum_{i=1}^{B} p_i(t) \sum_{j=1}^{B} \ell_j(\boldsymbol{\theta}(t)) - \sum_{i=1}^{B} \ell_i(\boldsymbol{\theta}(t)) \sum_{j=1}^{B} p_j(t)$$

$$= 2B \sum_{i=1}^{B} p_i(t)\ell_i(\boldsymbol{\theta}(t)) - 2 \sum_{i=1}^{B} p_i(t) \sum_{i=1}^{B} \ell_i(\boldsymbol{\theta}(t)) = 2B \sum_{i=1}^{B} p_i(t)\ell_i(\boldsymbol{\theta}(t)) - 2 \sum_{i=1}^{B} \ell_i(\boldsymbol{\theta}(t)),$$

$$\text{(B.46)}$$

which gives

$$\frac{1}{B} \sum_{i=1}^{B} \ell_i(\boldsymbol{\theta}(t)) \leq \sum_{i=1}^{B} p_i(t)\ell_i(\boldsymbol{\theta}(t)). \quad \text{(B.47)}$$

Hence, by the fact that $\hat{L}_n(\boldsymbol{\theta}^*) = 0 \Leftrightarrow \ell_i(\boldsymbol{\theta}^*) = 0, \forall i \in [n]$, we get

$$R(t) = \mathbb{E} \left[ \sum_{j \in \mathcal{B}_t} p_j(t)\ell_j(\boldsymbol{\theta}(t)) - \frac{1}{B} \sum_{j \in \mathcal{B}_t} \ell_j(\boldsymbol{\theta}(t)) \right] \geq 0, \quad \text{(B.48)}$$

which completes the proof. $\qquad \square$

---

[8]This means the order consistency: When one sample's loss is larger/smaller than that of the other, so does its weight.

**Remark 3.** *We emphasize again that the the order consistency Eq. (B.45) is a* sufficient *condition:* $\exists(\beta_1, \beta_2)$ *s.t. Eq. (B.45) holds* $\Rightarrow R(t) \geq 0 \Rightarrow \frac{1}{T}\sum_{t=0}^{T-1} R(t) \geq 0$, *while the reverse does not hold. That is, to guarantee* $\frac{1}{T}\sum_{t=0}^{T-1} R(t) \geq 0$, *one can include more betas, at least from the continuity of the sampling scheme Eq. (3.1) w.r.t. hyper-parameters.*

### B.4 PROOF OF THM. 3.3

*Proof.* Consider a continuous-time idealization of the sampling scheme Eq. (3.1):

$$s(t) = \beta_2 s(t-1) + (1-\beta_2)\ell(t) \Rightarrow s(t) - s(t-1) = (1-\beta_2)(\ell(t) - s(t-1)) \qquad \text{(B.49)}$$

$$\Rightarrow s'(t) = (1-\beta_2)(\ell(t) - s(t)), \qquad \text{(B.50)}$$

with $\ell(t) := \ell(\boldsymbol{\theta}(t))$, and $\beta_2 \neq 0$. Similarly,

$$w(t) = \beta_1 s(t-1) + (1-\beta_1)\ell(t) \Rightarrow w(t) - s(t) = (\beta_2 - \beta_1)(\ell(t) - s(t-1))$$

$$\Rightarrow w(t) = s(t) + (\beta_2 - \beta_1)\frac{s(t) - s(t-1)}{1 - \beta_2} \qquad \text{(by Eq. (B.49))}$$

$$\Rightarrow w(t) = s(t) + \frac{\beta_2 - \beta_1}{1 - \beta_2}s'(t)$$

$$\Rightarrow w(t) = (\beta_2 - \beta_1)\ell(t) + (1 - \beta_2 + \beta_1)s(t). \quad \text{(by Eq. (B.50))}$$
$$\text{(B.51)}$$

Since $\mathcal{L}\{\cdot\}$ is linear and satisfies $\mathcal{L}\{f'\}(\omega) = \omega\mathcal{L}\{f\}(\omega) - f(0)$, we have

$$\text{Eq. (B.50)} \Rightarrow \mathcal{L}\{s'\}(\omega) = (1-\beta_2)(\mathcal{L}\{\ell\}(\omega) - \mathcal{L}\{s\}(\omega)) = \omega\mathcal{L}\{s\}(\omega) - s(0)$$

$$\Rightarrow \mathcal{L}\{s\}(\omega) = \frac{1-\beta_2}{\omega + (1-\beta_2)}\mathcal{L}\{\ell\}(\omega) + \frac{s(0)}{\omega + (1-\beta_2)}, \qquad \text{(B.52)}$$

and

$$\text{Eq. (B.51)} \Rightarrow \mathcal{L}\{w\}(\omega) = (\beta_2 - \beta_1)\mathcal{L}\{\ell\}(\omega) + (1 - \beta_2 + \beta_1)\mathcal{L}\{s\}(\omega)$$

$$= \frac{(\beta_2 - \beta_1)\omega + (1-\beta_2)}{\omega + (1-\beta_2)}\mathcal{L}\{\ell\}(\omega) + \frac{(1 - \beta_2 + \beta_1)}{\omega + (1-\beta_2)}s(0), \quad \text{(by Eq. (B.52))}$$
$$\text{(B.53)}$$

$$= \frac{(\beta_2 - \beta_1)\omega + (1-\beta_2)}{\omega + (1-\beta_2)}\mathcal{L}\{\ell\}(\omega) + \mathcal{L}\{(1 - \beta_2 + \beta_1)s(0) \cdot e^{-(1-\beta_2)t}\}(\omega)$$

$$= \frac{(\beta_2 - \beta_1)\omega + (1-\beta_2)}{\omega + (1-\beta_2)}\mathcal{L}\{\ell\}(\omega) + O(1/n). \quad \text{(recall } s(0) = 1/n)$$
$$\text{(B.54)}$$

Then, the transfer function is $H(\omega) = \frac{(\beta_2-\beta_1)\omega+(1-\beta_2)}{\omega+(1-\beta_2)}$, with

$$|H(\mathrm{i}\omega_0)| = \left|\frac{(\beta_2 - \beta_1)\mathrm{i}\omega_0 + (1-\beta_2)}{\mathrm{i}\omega_0 + (1-\beta_2)}\right| = \sqrt{\frac{(\beta_2 - \beta_1)^2\omega_0^2 + (1-\beta_2)^2}{\omega_0^2 + (1-\beta_2)^2}}, \qquad \text{(B.55)}$$

and

$$|H(\mathrm{i}\omega_0)| \leq 1, \quad \lim_{\omega_0 \to +\infty} |H(\mathrm{i}\omega_0)| = |\beta_2 - \beta_1|. \qquad \text{(B.56)}$$

The proof is completed. $\qquad\qquad\qquad\qquad\qquad\qquad\qquad\qquad\qquad\qquad\square$

### B.5 ES TO SOLVE A DRO PROBLEM

From another perspective, ES can be also reformulated as a solution to a distributionally robust optimization (DRO) problem, or more specifically the minimax optimization problem

$$\min_{\boldsymbol{\theta}\in\Theta} \max_{\boldsymbol{p}\in\Delta^n} L_n(\boldsymbol{\theta}; \boldsymbol{p}) := \sum_{i=1}^{n} p_i(\ell_i(\boldsymbol{\theta}) - \ell_i^{\text{ref}}), \qquad \text{(B.57)}$$

where $\Delta^n$ denotes the $(n-1)$-dimensional probability simplex. This objective leads to a stronger requirement for robust performances on both typical and rare samples compared to the regular ERM (Shalev-Shwartz & Wexler (2016)). Different from traditional DRO, Eq. (B.57) introduces a reference loss $\ell_i^{\text{ref}}$, with the excess loss $\ell_i(\boldsymbol{\theta}) - \ell_i^{\text{ref}}$ measuring the improvement of the model on the $i$-th sample with respect to a reference model (typically *pre-trained*; see e.g. Oren et al. (2019); Xie et al. (2023a); Mindermann et al. (2022)). The second advantage of ES is to naturally leverage losses of historical models along the training dynamics as a proxy of the reference loss $\ell_i^{\text{ref}}$ in Eq. (B.57), which can be continuously updated without explicitly (pre-)training additional models.

Specifically, we have the following proposition.

**Proposition B.5.** *Consider to solve the minimax objective Eq. (B.57) via gradient ascent-descent*

$$\begin{cases} \boldsymbol{p}(t) \propto \boldsymbol{w}(t) := \boldsymbol{w}(t-1) + (1-\beta_1)(\boldsymbol{\ell}(\boldsymbol{\theta}(t)) - \boldsymbol{\ell}^{ref}(\boldsymbol{\theta}(1:t-1))), \\ \boldsymbol{\theta}(t+1) := \boldsymbol{\theta}(t) - \eta_t^{\boldsymbol{\theta}} \sum_{i=1}^n p_i(t) \nabla_{\boldsymbol{\theta}} \ell_i(\boldsymbol{\theta}(t)), \end{cases} \tag{B.58}$$

*where the reference loss is defined as $\boldsymbol{\ell}^{ref}(\boldsymbol{\theta}(1:t)) := [\ell_i^{ref}(\boldsymbol{\theta}(1:t))]_{i \in [n]}$ with $\ell_i^{ref}(\boldsymbol{\theta}(1:t)) := \frac{1-2\beta_1+\beta_1\beta_2}{1-\beta_1}\ell_i(\boldsymbol{\theta}(t)) + \frac{\beta_1(1-\beta_2)^2}{1-\beta_1}\sum_{k=1}^{t-1}\beta_2^{t-1-k}\ell_i(\boldsymbol{\theta}(k)) + \frac{\beta_1(1-\beta_2)\beta_2^{t-1}}{n(1-\beta_1)}, i \in [n]$. Then, the dynamics Eq. (B.58) is consistent with gradient descent sampled with the sampling scheme Eq. (3.1).*

*Proof.* The problem Eq. (B.57) can be solved in an alternative gradient descent-ascent manner:

$$\boldsymbol{\theta}(t+1) = \boldsymbol{\theta}(t) - \eta_t^{\boldsymbol{\theta}} \sum_{i=1}^n p_i(t) \nabla_{\boldsymbol{\theta}} \ell_i(\boldsymbol{\theta}(t)),$$

$$w_i(t+1) = w_i(t) + \eta_t^{\boldsymbol{w}}(\ell_i(\boldsymbol{\theta}(t+1)) - \ell_i^{\text{ref}}), \quad p_i(t) = \frac{w_i(t)}{\sum_j w_j(t)}. \tag{B.59}$$

The sampling scheme Eq. (3.1) updates the weights as

$$w_i(t+1) = w_i(t) + (1-\beta_1)(\ell_i(\boldsymbol{\theta}(t+1)) - \ell_i(\boldsymbol{\theta}(t))) + \beta_1(s_i(t) - s_i(t-1)). \tag{B.60}$$

By Eq. (B.19), we get

$$s_i(t) - s_i(t-1) = -(1-\beta_2)\beta_2^{t-1}s_i(0) - (1-\beta_2)^2 \sum_{k=1}^{t-1}\beta_2^{t-1-k}\ell_i(\boldsymbol{\theta}(k)) + (1-\beta_2)\ell_i(\boldsymbol{\theta}(t)),$$

hence

$$w_i(t+1) = w_i(t) + (1-\beta_1)(\ell_i(\boldsymbol{\theta}(t+1)) - \ell_i(\boldsymbol{\theta}(t))) - \beta_1(1-\beta_2)\beta_2^{t-1}s_i(0)$$

$$- \beta_1(1-\beta_2)^2 \sum_{k=1}^{t-1}\beta_2^{t-1-k}\ell_i(\boldsymbol{\theta}(k)) + \beta_1(1-\beta_2)\ell_i(\boldsymbol{\theta}(t)). \tag{B.61}$$

Let

$$\ell_i^{\text{ref}} = \frac{1-2\beta_1+\beta_1\beta_2}{1-\beta_1}\ell_i(\boldsymbol{\theta}(t)) + \frac{\beta_1(1-\beta_2)^2}{1-\beta_1}\sum_{k=1}^{t-1}\beta_2^{t-1-k}\ell_i(\boldsymbol{\theta}(k)) + \frac{\beta_1(1-\beta_2)\beta_2^{t-1}}{1-\beta_1}s_i(0), \tag{B.62}$$

then we have

$$w_i(t+1) = w_i(t) + (1-\beta_1)(\ell_i(\boldsymbol{\theta}(t+1)) - \ell_i^{\text{ref}}), \tag{B.63}$$

which coincides with the update formula Eq. (B.59) with $\eta_t^{\boldsymbol{w}} = 1 - \beta_1$. The proof is completed. $\square$

## C   MORE DETAILS OF ALGORITHMS

This section presents more details of the ES(WP) sampling framework.

**Annealing (optional)** Notably, similar to the loss-weighted sampling scheme Eq. (2.3) and its further variants, the sampling scheme Eq. (3.1) also assigns different weights on the respective gradient of data samples, leading to a biased estimation on the true gradient $\nabla_{\boldsymbol{\theta}} \hat{L}_n(\cdot)$ (with uniform individual weights). Inspired by Qin et al. (2024), we adopt the *annealing* strategy, to perform normal training (with the standard batched sampling, no data selection) at the last few epochs. Besides, to get better initializations of the weights $\{w_i(\cdot)\}_{i\in[n]}$, we also apply the annealing strategy at the first few epochs.

Combining the sampling scheme Eq. (3.1) with the annealing strategy, we obtain the **Evolved Sampling** (**ES**) framework (formalized in Alg. 1).

---

**Algorithm 1 E**volved **S**ampling (**W**ith **P**runing)

---

**Require:** Dataset $\mathcal{D} = \{\boldsymbol{z}_i\}_{i=1}^n$, optimizer (e.g. Adam)
**Require:** Pruning ratio $r$, meta-batch size $B$, mini-batch size $b \leq B$, total epochs $E$, annealing
   epochs $(E_{a_{\text{start}}}, E_{a_{\text{end}}})$, hyper-parameters $\beta_1, \beta_2 \in (0, 1)$
   Initialize the scores/weights $\boldsymbol{s}(0) = \boldsymbol{w}(0) = \frac{1}{n}\mathbf{1}_n, t = 0$
   **for** $e = 0, 1, \cdots, E - 1$ **do**
      **if** $E_{a_{\text{start}}} \leq e < E - E_{a_{\text{end}}}$ **then**
         Sample a sub-dataset $\mathcal{D}_e$ ($|\mathcal{D}_e| = (1-r)|\mathcal{D}|$) from $\mathcal{D}$ without replacement, according to the
         probability $p_i'(e) \propto w_i(e)$                                          ▷ *pruning*
      **else**
         Set $\mathcal{D}_e = \mathcal{D}$
      **end if**
      **for** $j = 0, 1, \cdots, \lceil \frac{|\mathcal{D}_e|}{B} \rceil - 1$ **do**
         Sample a meta-batch $\mathcal{B}_t$ ($|\mathcal{B}_t| = B$) uniformly from $\mathcal{D}_e$ without replacement
         Compute the loss $\ell_i(\boldsymbol{\theta}(t))$ for $\boldsymbol{z}_i \in \mathcal{B}_t$
         Update score: $s_i(e+1) \leftarrow \beta_2 s_i(e) + (1-\beta_2)\ell_i(\boldsymbol{\theta}(t))$ for $\boldsymbol{z}_i \in \mathcal{B}_t$
         Update the weight: $w_i(e+1) \leftarrow \beta_1 s_i(e) + (1-\beta_1)\ell_i(\boldsymbol{\theta}(t))$ for $\boldsymbol{z}_i \in \mathcal{B}_t$
         **if** $E_{a_{\text{start}}} \leq e < E - E_{a_{\text{end}}}$ **then**
            Sampling a mini-batch $\mathfrak{b}_t$ ($|\mathfrak{b}_t| = b$) from $\mathcal{B}_t$ without replacement, according to the
            probability $p_i(e+1) \propto w_i(e+1)$
            Update model: $\boldsymbol{\theta}(t+1) \leftarrow \text{optimizer}(\boldsymbol{\theta}(t); \mathfrak{b}_t)$
         **else**
            Update model: $\boldsymbol{\theta}(t+1) \leftarrow \text{optimizer}(\boldsymbol{\theta}(t); \mathcal{B}_t)$          ▷ *annealing*
         **end if**
         $t \leftarrow t + 1$
      **end for**
   **end for**

---

**Pruning (optional)** Note that applying the sampling scheme Eq. (3.1) to meta-batches (with the batch size $B$) in fact introduces data selection in a *batch* level, since one can always select a smaller batch (with the batch size $b < B$) out of the meta-batch, according to the sampling probability $p_i(t)$ defined in Eq. (3.1). For more aggressive data pruning and enhanced data efficiency, we can further extend ES by involving the *set* level data selection. That is, randomly pruning the *whole* dataset according to the probability proportional to the weights $\{w_i(e)\}_{i=1}^n$ at the beginning of the $e$-th epoch. This is formalized as **Evolved Sampling with Pruning** (**ESWP**) in Alg. 1.

# D  MORE DETAILS OF EXPERIMENTS

In this section, we present further experimental results and details. We run all the experiments with one NVIDIA A100 (80GB) with the mixed-precision training except the pre-training of ViT-Large on ImageNet-1K. All the algorithms are implemented based on PyTorch (Paszke et al. (2019)) and Timm (Wightman et al. (2019)). For InfoBatch, our implementation is adapted from Qin et al. (2024).

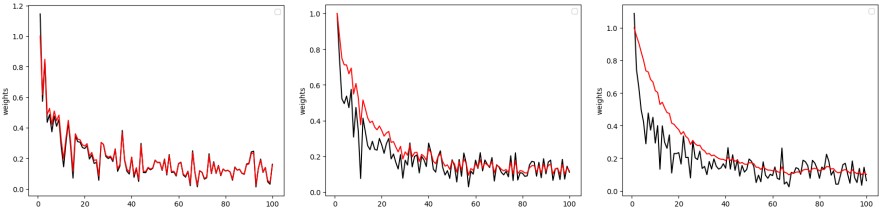

Figure 8: The output weights of different sampling schemes, where the black curves denote Eq. (2.3), while the red curves represent Eq. (3.1) (from left to right: $\beta_1 = 0.1, 0.5, 0.8$, and $\beta_2 \equiv 0.9$). Here, we draw the black curve as a decayed function with random perturbations, to mimic typical behaviors of loss curves in general machine learning tasks. It is shown that the sampling scheme Eq. (2.3) is usually sensitive w.r.t. oscillations. However, when losses oscillate, the sampling scheme Eq. (3.1) reacts moderately by not only reserving some portion of dynamical details of losses (high frequencies), but also remaining necessary robustness by capturing the overall trend (low frequencies), with the flexibility to trade off in between by tuning $(\beta_1, \beta_2)$.

## D.1 ILLUSTRATIONS ON SYNTHETIC DATASETS

## D.2 SELECTED SAMPLES BY ES(WP)

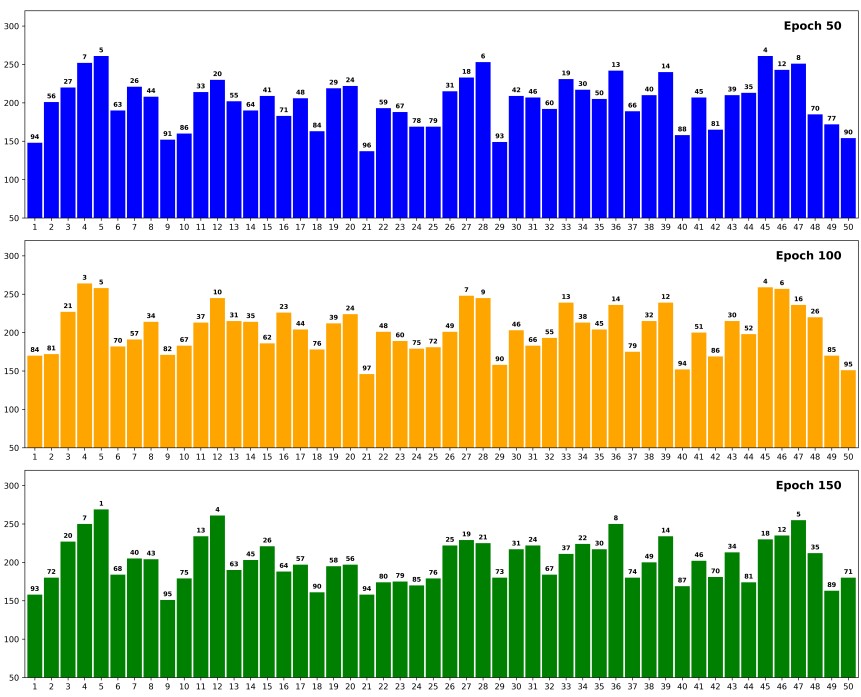

Figure 9: Visualization of the number of selected samples for BP of each class in ESWP (ResNet-50, Cifar-100), following Figure 6 in Thao Nguyen et al. (2023). Here, it shows the result of the first 50 classes. The number on top of each column shows the rank over 100 classes (a lower rank indicates a higher number of selected samples). It is shown that ES(WP) can automatically adjust selected samples at different training stages.

## D.3 EXPERIMENTS ON CIFAR DATASETS

For computer vision (CV) tasks, we train ResNet-18/50 (R-18/50) models on CIFAR-10/100 datasets, using SGD for 200 epochs, with $B = 128/256$ for ResNet-18/50 ($b/B = 50\%$ for ResNet-50).

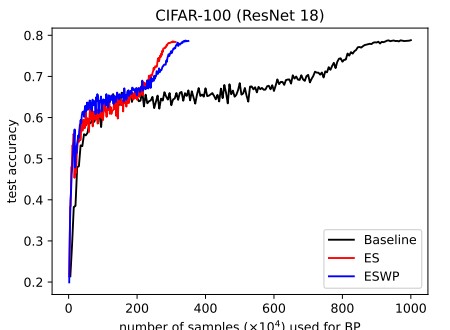 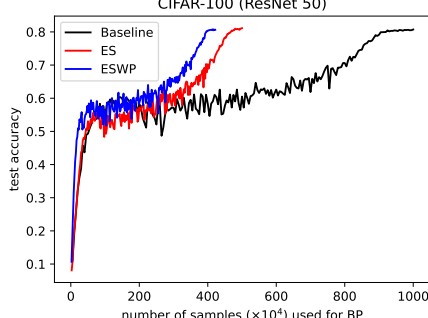

Figure 10: Learning dynamics of different data selection methods: Test accuracy versus the number of samples used for back propagations (BPs).

For the experiments on the CIFAR-10/100 datasets, we use the SGD optimizer with the momentum $0.9$ and weight decay $5 \times 10^{-4}$. We apply the OneCycle scheduler (Smith & Topin (2019)) with the cosine annealing. For CIFAR-10, the maximal learning rate is $0.2$ for the baseline and *set* level selection methods, while $0.05$ for *batch* level selection methods due to larger variances of stochastic gradients and $0.08$ for ESWP. For CIAFR-100 trained with ResNet-18/50, the maximal learning rates for all the sampling methods are $0.05/0.2$, following Qin et al. (2024).

### D.4 EXPERIMENTS OF FULL FINE-TUNING

**Vision Transformer.** We fine-tune ViT-Large model on ImageNet-1K with a meta-batch size $B = 256$ for 10 epochs, using the Adam optimizer with the OneCycle scheduler (Smith & Topin (2019)) with the cosine annealing and a maximal learning rate of $2 \times 10^{-5}$.

**ALBERT.** Following the setup in Xie et al. (2023b) (Table 8), we use the AdamW optimizer and the polynomial decay scheduler with warm up.

### D.5 EXPERIMENTS OF PRE-TRAINING

We conduct the MAE-based pre-training of ViT-Large on ImageNet-1K using $4\times$A100 GPUs. Following the setup in He et al. (2022), we train for 300 epochs with a 40-epoch warmup, base learning rate $1.5 \times 10^{-4}$, weight decay $0.05$, and batch sizes $(B, b) = (256, 256)$ per GPU for ESWP, i.e., there is no batch level data selection. In our implementation, the sampling procedure of ESWP is conducted by an additional round of synchronization.

After pre-training, we fine-tune the model for 50 epochs with a 5-epoch warmup, using the standard batched sampling (no data selection) with the batch size $B = 256$ per GPU.

### D.6 EXPERIMENTS ON FINE-TUNING QWEN

**Training Details** We conduct experiments on a single A100 (40GB) GPU to investigate the low-resource regime. Our implementation builds upon the verl framework.[9] We set the batch sizes $B = 32, b = b_{\text{micro}} = 8$, and use the AdamW optimizer with a learning rate of $1 \times 10^{-5}$, which follows a cosine decay scheduler with a warm-up ratio of $0.1$. We set the total epoch as 10 and evaluate the model after 1K, 2K, and 4K training steps.

**Evaluation Details** The detailed breakdown of pass@1 results are shown in Tab. 9. We use a temperature of $1.0$, top_p=1, the default chat template and Chain-of-Thought (CoT) prompting for evaluation.

---

[9]https://github.com/volcengine/verl

Table 9: Pass@1 accuracy on MATH500, AIME24, and Olympiad Bench under different training budgets.

| Method (Steps, Time) | MATH500 | AIME24 | Olympiad Bench | Averaged |
|---|---|---|---|---|
| Baseline (1K, 50min) | **61.8** | 6.7 | 26.2 | 31.6 |
| Baseline (2K, 100min) | 59.6 | **10.0** | 27.7 | 32.4 |
| Baseline (4K, 200min) | 63.4 | 13.3 | 25.2 | 34.0 |
| ESWP (1K, 26.5min) | **61.8** | **10.0** | **27.4** | **33.1** |
| ESWP (2K, 53min) | **65.2** | **10.0** | **28.6** | **34.6** |
| ESWP (4K, 106min) | **65.6** | **16.7** | **32.1** | **38.1** |

## D.7 COMPARISON METHODS: DEFAULT HYPER-PARAMETERS

For all the other data selection methods, we also use their default hyper-parameters in original papers in our experiments. Therefore, the comparisons and evaluations are fair in terms of hyper-parameters. We list the default hyper-parameters of all the other data selection methods as follows:

- InfoBatch (Qin et al. (2024)): pruning ratio $r = 0.5$, annealing ratio $1 - \delta = 0.125$;
- KAKURENBO (Thao Nguyen et al. (2023)): pruning ratio $r = 0.3$, confidence threshold $\tau = 0.7$;
- UCB (Raju et al. (2021)): pruning ratio $r = 0.3$, decay parameter $\beta = 0.8$, confidence bound $c = 1$;
- Loss (Katharopoulos & Fleuret (2017)), Order (Kawaguchi & Lu (2020)): the same batch sizes as ES.

## THE USE OF LLMS

This work uses LLMs only to confirm the correct usage of English words and phases.

