# OpenReview forum: "Data-Efficient Training by Evolved Sampling"
_ICLR.cc/2026/Conference — Submitted to ICLR 2026_

### Official Review · Reviewer_evuj · 2025-10-28

**Soundness:** 3
**Presentation:** 2
**Contribution:** 3
**Rating:** 6
**Confidence:** 3

**Summary:**

The paper proposes to identify informative data samples in order to reduce back propagation time. The proposed method Evolved Sampling (EV), as well as ES with Pruning (ESWP) determines the importance/weights of data samples based on both (zero-order) losses and additional (first-order) loss differences along the training dynamics. Results show lossless training, saving up to 45% wall-clock time.

**Strengths:**

The paper is original in combining batch-level and set-level sampling.

Results are extensive, with gains in low-resource settings and larger acceleration in LLM fine-tuning beyond CIFAR-10/100.

A Fourier-based analysis explains the observed benefits.

**Weaknesses:**

Lack of clarity: see Questions on which points are not clear

No variance: it is not clear how the measurements vary and how statistically significant are the results.

Profiling: paper could benefit from supporting claims on computational and memory efficiency in the main text with numbers.

For other Questions please refer to the Questions.

**Questions:**

1. Please clarify whether the reported results are means over multiple seeds or single runs; in close comparisons, multi-seed averages can be lower and differences may lack statistical significance relative to the baselines.

2. In Table 1, shouldn’t the percentage of samples for BP rather than using the “#” symbol, which suggests a raw count and is confusing?

3. Is convexity a reasonable assumption in the proof, and can the result be established under milder conditions?

4. Please elaborate on computational efficiency at line 278; the claim that overhead is modest because FP uses far fewer FLOPs than BP needs exact FLOP counts and wall-clock timings, since BP is typically about twice the cost of FP and not negligible.

5. For the memory efficiency claim after Eq. 3.1, please provide concrete numbers showing that storing per-sample scores or weights is negligible relative to the data tensor footprint.

6. Since the beta parameters require a grid search, please report the end-to-end wall-clock time to select beta and then train in a practical scenario, or should one use (0.2, 0.8) across datasets every time?

7. Do the beta values interact with other hyperparameters such as the learning rate, or can standard hyperparameters be reused without retuning?

8. Could you provide intuition for why ESWP outperforms ES in Table 3?

9. At line 375, please specify which components are distributed and how.

10. For Table 8, describe how Random is implemented, how the learning rate is handled under Random sampling, and whether the randomness is static or dynamic across iterations.
Please also explain how methods that add selection overhead can surpass Random in efficiency, given that Random should incur the least training overhead.

---

> ### Author Response · Authors · 2025-11-24
> **Response to Reviewer evuj (1st part)**
>
> > **Q1**: "Please clarify whether the reported results are means over multiple seeds or single runs; in close comparisons, multi-seed averages can be lower and differences may lack statistical significance relative to the baselines."
>
> **A1**: As stated in Line 364-365, all reported results are evaluated on the average of 3-4 independent random trials. We did not include corresponding variances in the original manuscript for compactness. As required, here we report numbers of standard deviations of as many experiments in **Table 1'**.
>
> - **Table 1'**: Test accuracies of different data selection methods ($\text{mean}\pm \text{std}$).
>
>     ||Baseline|Order|Loss|Infobatch|UCB|KAKURENBO|ES|ESWP
>     |----|----|----|----|----|----|----|----|----
>     |CIFAR-10 (ResNet-18)|$95.4_{\pm 0.0}$|$95.4_{\pm 0.2}$|$95.3_{\pm 0.1}$|$95.3_{\pm 0.1}$|$95.2_{\pm 0.1}$|$95.3_{\pm 0.1}$|$95.4_{\pm 0.1}$|$95.3_{\pm 0.0}$
>     |CIFAR-100 (ResNet-18)|$78.8_{\pm 0.0}$|$78.5_{\pm 0.0}$|$78.4_{\pm 0.1}$|$78.4_{\pm 0.1}$|$77.6_{\pm 0.1}$|$78.1_{\pm 0.2}$|$78.8_{\pm 0.0}$|$78.6_{\pm 0.1}$
>     |ImageNet-1K (ViT-Large)|$84.4_{\pm 0.1}$|$84.2_{\pm 0.2}$|$84.3_{\pm 0.1}$|$84.7_{\pm 0.1}$|$84.2_{\pm 0.2}$|$84.3_{\pm 0.2}$|$84.7_{\pm 0.1}$|$85.0_{\pm 0.1}$
>
> > **Q2**: "In Table 1, shouldn’t the percentage of samples for BP rather than using the “#” symbol, which suggests a raw count and is confusing?"
>
> **A2**: Thanks for pointing it out. We have fixed this.
>
> > **Q3**: "Is convexity a reasonable assumption in the proof, and can the result be established under milder conditions?"
>
> **A3**: We have added discussions on the convexity assumption in Remark 1 (Point 1). In fact, the convexity assumption is widely-adopted and also standard in proving the convergence of (S)GD in optimization literature (e.g. [1]). Under the non-convex setting, typically only the *convergence to stationary points* can be guaranteed for general functions (see e.g. [2]).
>
> > **Q4**: "Please elaborate on computational efficiency at line 278; the claim that overhead is modest because FP uses far fewer FLOPs than BP needs exact FLOP counts and wall-clock timings, since BP is typically about twice the cost of FP and not negligible."
>
> **A4**: Computing the exact FLOPs of back propagations is technically challenging, and is typically approximated as roughly twice of that in forward passes (see e.g. the DeepSpeed FLOPs profiler). Nevertheless, we report the following wall-clock time count, which directly measures the efficiency gains achieved by ESWP.
>
> - **Table 2'**: The wall-clock time ($s$) of FP/BP within an epoch when training ResNet-50 on CIFAR-100 using a RTX 3090.
>
> |      | Forward | Backward | Total |
> |------|-------|-------|-------|
> | Baseline ($B=256$) | $1.7$  | $15.5$    | $17.2$ |
> | ESWP ($b=128$, $r=0.2$) | $4.6$    | $6.5$     | $11.1$    |
>
> Here, the forward time of ESWP includes all required overheads, such as re-computing mini-batch losses and updating sample scores/weights. As shown in **Table 2'**, these overheads are less significant relative to the substantial reduction in back propagations.
>
> > **Q5**: "For the memory efficiency claim after Eq. 3.1, please provide concrete numbers showing that storing per-sample scores or weights is negligible relative to the data tensor footprint."
>
> **A5**: As stated in Line 412-414, in the experiment of fully fine-tuning ViT-Large on ImageNet-1K, "We also numerically evaluate the corresponding overall memory usage of ES (49.7GB) and ESWP (49.1GB), which are also reduced compared to Baseline (52.4GB), verifying the efficiency of ES(WP) in terms of memory loads."
>
> > **Q6**: "Since the beta parameters require a grid search, please report the end-to-end wall-clock time to select beta and then train in a practical scenario, or should one use (0.2, 0.8) across datasets every time?"
>
> **A6**: We have outlined the hyper-parameters tuning aspect in Section 3.4 in the original manuscript. In experiments, we take the default values of $(\beta_1, \beta_2)$, which are obtained by a fine-grained grid search in small-scale simulations (Section 4.3, Paragraph "Choices of $(\beta_1,\beta_2)$"). These defaults are consistently validated to be (locally) optimal in small-scale experiments (Figure 6 and Figure 7), and their superior effectiveness remains on large-scale tasks (Section 4.1, (ii) \& (iii) and Section 4.2). This is *a common routine* of  hyper-parameters tuning, which is also adopted in e.g. [3, 4, 5], to reuse default hyper-parameters (obtained by grid search in small-scale simulations) in large-scale experiments, without further tuning.

---

> ### Author Response · Authors · 2025-11-24
> **Response to Reviewer evuj (2nd part)**
>
> > **Q7**: "Do the beta values interact with other hyperparameters such as the learning rate, or can standard hyperparameters be reused without retuning?"
>
> **A7**: The default beta values are reused without retuning, and do not interact with other hyper-parameters such as the learning rate.
>
> > **Q8**: "Could you provide intuition for why ESWP outperforms ES in Table 3?"
>
> **A8**: The intuition is, since the ViT-Large model has been pre-trained on the ImageNet-21K dataset, it has already learned potential data features. Hence, we do not need all data samples to avoid the risk of over-fitting, implying the benefit of data pruning.
>
> > **Q9**: "At line 375, please specify which components are distributed and how."
>
> **A9**: We use pytorch DDP to distribute the original large dataset to different GPU nodes.
>
> > **Q10**: "For Table 8, describe how Random is implemented, how the learning rate is handled under Random sampling, and whether the randomness is static or dynamic across iterations. Please also explain how methods that add selection overhead can surpass Random in efficiency, given that Random should incur the least training overhead."
>
> **A10**: Random is implemented following [3] (Table 1), which is dynamic across iterations and on the set level, in the sense that we uniformly prune data samples according to a given ratio at every training epoch. The learning rate is set as the same as Baseline and other set level data selection methods. Despite of simplicity, Random does not use the model-data interaction information along the training process, potentially leading to decelerated convergence and hence worse overall efficiency.
>
>
> **References**
>
> [1] Guillaume Garrigos and Robert M. Gower. Handbook of convergence theorems for (stochastic) gradient methods. *arXiv preprint arXiv:2301.11235*, 2023.
>
> [2] Ahmed Khaled and Peter Richtarik. Better theory for SGD in the nonconvex world. *Transactions on Machine Learning Research*, 2023.
>
> [3] Ziheng Qin, Kai Wang, Zangwei Zheng, Jianyang Gu, Xiangyu Peng, Zhaopan Xu, Daquan Zhou, Lei Shang, Baigui Sun, Xuansong Xie, and Yang You. InfoBatch: Lossless training speed up by unbiased dynamic data pruning. In *International Conference on Learning Representations*, 2024.
>
> [4] Ziteng Wang, Jianfei Chen, and Jun Zhu. Efficient backpropagation with variance controlled adaptive sampling. In *International Conference on Learning Representations*, 2024.
>
> [5] Truong Thao Nguyen, Balazs Gerofi, Edgar Josafat Martinez-Noriega, Francois Trahay, and Mohamed Wahib. KAKURENBO: Adaptively hiding samples in deep neural network training. *Advances in Neural Information Processing Systems*, volume 36, pp. 37900–37922, 2023.

---

### Official Review · Reviewer_DNpN · 2025-10-31

**Soundness:** 3
**Presentation:** 3
**Contribution:** 2
**Rating:** 4
**Confidence:** 4

**Summary:**

The paper proposes "Evolved Sampling (ES)", a dynamic sampling method that employs an exponential moving average scheme to select training samples based on loss values during training. It reduces the number of samples required for backpropagation and shortens the necessary training time.

**Strengths:**

1. The paper provides a comprehensive set of experiments covering small-scale (CIFAR), large-scale (ImageNet, ViT-Large), NLP (GLUE), and even large language model fine-tuning tasks. This demonstrates the robustness and general applicability of the method.

2. Including thorough ablation studies (e.g., the importance of loss differences, annealing, pruning strategies, and hyperparameter sensitivity ).

**Weaknesses:**

​1. The paper lacks a systematic comparison with a random sampling baseline. For example, in experimental results such as Table 2 and Table 3, a baseline that purely randomly selects data subsets is absent.

2. ​Limited novelty, the core innovation of the method essentially introduces an "exponential moving average with historical score" mechanism into dynamic sampling.


3.  The weight update formula (Equation 3.1) is somewhat heuristic, primarily relying on an exponential moving average mechanism. It lacks a solid theoretical derivation.

4. The assumption in Proposition 2 that "there exists $\theta^{\star}$ such that  $\hat{L}_n(\theta^{*}) = 0$... "  is too strong for me.

**Questions:**

1. Why is the specific formulation of Eq. (3.1), which incorporates the difference between the current loss and the historical trajectory (as shown in Proposition 3.1), more beneficial than simply using the smoothed historical score si​(t) for weighting?  Can you provide a more detailed comparison, including a simple hyperparameter search for $s_i(t)$， to illustrate the necessity of your method？

2. Even if the stability of sampling is resolved, why is the sampling scheme in Eq. (3.1) helpful for training？

3. How sensitive is the performance of ES(WP) to deviations from the default (β1​,β2​)values on different tasks? Are there any heuristics or correlations between optimal ($\beta_1$, $\beta_2$​)and dataset properties?

4. The validity of Proposition 2.1 under standard training regimes appears questionable. It would be valuable to empirically validate it on synthetic data, and, if feasible, to explore its robustness under relaxed assumptions.

---

> ### Author Response · Authors · 2025-11-24
> **Response to Reviewer DNpN (1st part)**
>
> > **W1**: "The paper lacks a systematic comparison with a random sampling baseline. For example, in experimental results such as Table 2 and Table 3, a baseline that purely randomly selects data subsets is absent."
>
> **A1**: We report the results of purely random sampling (Random) in Table 2 and Table 3 in the updated manuscript, with the same data pruning ratio and other hyper-parameters as ESWP. We observe that Random consistently underperforms ESWP in Table 2, 3 and 7.
>
> > **W2**: "Limited novelty, the core innovation of the method essentially introduces an "exponential moving average with historical score" mechanism into dynamic sampling."
>
> **A2**: With respect, we kindly point out that it is totally an oversimplification to view ES as common EMA.
> - In theory: ES is a non-trivial extension of EMA.
>     - In formulation, ES reduces to EMA by simply setting $\beta_1=\beta_2$.
>     - As shown in **Eq. (3.3)** in **Proposition 3.1**, despite that there are calculations regarding only losses in ES (Eq. (3.1)), Eq. (3.1) also *implicitly* leverages additional **correlations between gradients** to determine sample weights (details in Line 196-206), and this augmented high-order gradient (correlation) information has the coefficient $(\beta_2-\beta_1)$. That is, although both ES and EMA use losses to compute sample weights, the dynamic design of ES enables it to additionally (and implicitly) exploit gradient information along the training, while EMA can not. Clearly, different values of two betas also make it possible for ES to flexibly tune the strengths of losses and gradients when computing sample weights.
>     - As discussed in Line 216-219 and Footnote 3, ES can take the importance of both current losses and historical weights into account by tuning betas: By setting $(\beta_1, \beta_2) \to (0^+, 1^-)$, we are able to exploit the long-term historical information during training (via $\beta_2$), while simultaneously responding to current losses (via $\beta_1$). This is impossible for EMA.
> - In experiments: The superiority of ES over EMA is consistently verified via both ablation studies and real-world experiments.
>     - As shown in Table 6 (the last two rows) and recapped in Line 461-463 in the original manuscript, ES ("Dif") consistently outperforms EMA ("NonDif") on both image and language datasets.
>     - As shown in Table 2 and Table 3 in the original manuscript, ES(WP) consistently outperforms UCB ([4], which is exactly a type of  EMA-based sampling) for both small-scale and large-scale tasks.
>
> > **W3**: "The weight update formula (Equation 3.1) is somewhat heuristic, primarily relying on an exponential moving average mechanism. It lacks a solid theoretical derivation."
>
> **A3**: We update **Proposition 3.1** for theoretical justifications of the superiority of ES over EMA. See Point 2 under Point "In theory" in **A2**. The main insight is, although ES computes the sample weights with comparable calculations to EMA (since only losses are used for both ES and EMA, see Eq. (3.1)), ES can  implicitly exploit additional gradient correlation information along the training to determine sample weights.
>
> > **W4**: "The assumption in Proposition 2 that "there exists $\theta^*$ such that $\hat{L}_n (\theta^*)=0$..." is too strong for me."
>
> **A4**: We have demonstrated this assumption in Footnote 6 (and also in Point 2 in Remark 1). This condition simply requires that models can be trained with zero losses, which is empirically supported in e.g. [1] (Figure 1 (a)).
>
> > **Q1**: "Why is the specific formulation of Eq. (3.1), which incorporates the difference between the current loss and the historical trajectory (as shown in Proposition 3.1), more beneficial than simply using the smoothed historical score $s_i(t)$ for weighting? Can you provide a more detailed comparison, including a simple hyperparameter search for $s_i(t)$, to illustrate the necessity of your method?"
>
> **A5**: Please see **A2** and **A3** for details.
>
> > **Q2**: "Even if the stability of sampling is resolved, why is the sampling scheme in Eq. (3.1) helpful for training？"
>
> **A6**: Please see **A2** and **A3** for details.
>
> > **Q3**: "How sensitive is the performance of ES(WP) to deviations from the default $(\beta_1, \beta_2)$ values on different tasks? Are there any heuristics or correlations between optimal $(\beta_1, \beta_2)$ and dataset properties?"
>
> **A7**: For multiple datasets that we studied in this work, the default $(\beta_1, \beta_2)$ values are consistently optimal and robust (see Figure 6 and Figure 7), and work well on many other (large-scale) tasks. Regarding the deviations from defaults, it seems that given the same grid size of betas, the performance drops a little more for more challenging CIFAR-100 (by comparing Figure 6 (a) and Figure 6(b)), and so does it for CoLA with more conceptual subtlety (by comparing Figure 6 (a, b) and Figure 6(c)), despite that the defaults are consistently optimal.

---

> ### Author Response · Authors · 2025-11-24
> **Response to Reviewer DNpN (2nd part)**
>
> > **Q4**: "The validity of Proposition 2.1 under standard training regimes appears questionable. It would be valuable to empirically validate it on synthetic data, and, if feasible, to explore its robustness under relaxed assumptions."
>
> **A8**: The superiority of loss-weighted GD over vanilla GD can be verified in practical real-world simulations (e.g. **Table 1***), where the assumptions of Proposition 2.1 do not hold.
>
> **Table 1***: Test losses of ResNet-18 on CIFAR-10 at late training stages.
>
> | Epoch              | $150$   | $160$   | $170$   | $180$   | $190$   | $200$   |
> |--------------------|-------|-------|-------|-------|-------|-------|
> | Vanilla GD           | $0.342$ | $0.301$ | $0.280$ | $0.252$ | $0.239$ | $0.237$ |
> | Loss-Weighted GD | $0.304$ | $0.297$ | $0.270$ | $0.242$ | $0.234$ | $0.234$ |
>
>
> **References**
>
> [1] Chiyuan Zhang, Samy Bengio, Moritz Hardt, Benjamin Recht, and Oriol Vinyals. Understanding deep learning requires rethinking generalization. In *International Conference on Learning Representations*, 2017.

---

### Official Review · Reviewer_qPut · 2025-11-01

**Soundness:** 3
**Presentation:** 3
**Contribution:** 2
**Rating:** 6
**Confidence:** 5

**Summary:**

This work proposes to enhance dynamic data pruning by conducting batch-level data selection based on the dynamics of losses and augmented loss differences. It demonstrates an improved lossless saving ratio on the Transformer architectures.

**Strengths:**

1. The theoretical motivation is reasonable and meaningful. Turning it into a pruning probability captured the key benefits of dynamic data pruning.
2. The method design, probably inspired by the moving average in optimizers, is quite clever and efficient. It captures the losses’ dynamical differences without introducing any memory space dependence on epoch number. Compared to the uncertainty term in UCB, this design is more reasonable and is shown to be effective.
3. On Transformer architectures, the proposed method demonstrates improved data efficiency.
4. This work extends evaluation to more language model tasks and demonstrates the effectiveness.

**Weaknesses:**

1. The Tab.2 baseline performance of ResNet on the CIFAR datasets seems to be lower than generally reported. So does the saving ratio of the baseline method InfoBatch.
2. As there is no claim on code availability, the current experiment details could be insufficient. See questions.

**Questions:**

1. For Tab.3 finetuning ViT-Large on ImageNet-1K, which pretraining dataset does ViT-Large use?

---

> ### Author Response · Authors · 2025-11-24
> **Response to Reviewer qPut**
>
> > **W1**: "The Tab.2 baseline performance of ResNet on the CIFAR datasets seems to be lower than generally reported. So does the saving ratio of the baseline method InfoBatch."
>
> **A1**: While the test accuracy on CIFAR-10 (ResNet-18) here is slightly lower than the numbers reported in Table 1 of InfoBatch ($95.3$ v.s. $95.6$), our reported results on CIFAR-100 (ResNet-18) are higher ($78.4$ v.s. $78.2$). These discrepancies are minor and mainly attributed to randomness. For the saving ratio, we report the reduction in wall-clock training time to emphasize accelerations in practice; while InfoBatch reports the pruning ratio of datasets, which are not identical to practical time savings due to additional computation overheads.
>
> > **W2**: "As there is no claim on code availability, the current experiment details could be insufficient. See questions."
>
> **A2**: We have added the corresponding statement in Footnote 5.
>
> > **Q1**: "For Tab.3 finetuning ViT-Large on ImageNet-1K, which pretraining dataset does ViT-Large use?"
>
> **A3**: The ViT-Large model is pre-trained on the ImageNet-21K dataset.

---

### Official Review · Reviewer_WM6S · 2025-11-03

**Soundness:** 3
**Presentation:** 2
**Contribution:** 2
**Rating:** 2
**Confidence:** 4

**Summary:**

This paper proposes a dynamic data sampling framework called Evolved Sampling (ES) and its pruned version (ESWP). The goal is to accelerate model training through data selection while preserving model performance. The core of the method is to evaluate data sample importance based on the dynamics of both (zero-order) losses and (first-order) loss differences.

**Strengths:**

1. The primary strength of this paper is its extensive experimental validation.
2. "Lossless" Acceleration:  these speed-ups are achieved without a drop in model performance.
3. As a loss-based method, the framework is "much more light-weighted" than gradient-based sampling approaches.

**Weaknesses:**

1. Proposition 2.1 analyzes loss-weighted gradient descent, which corresponds to the baseline method "Loss" (Eq. 2.3), not the core ES algorithm proposed by the authors (Eq. 3.1).
2. The Fourier analysis in Theorem 3.2 merely concludes that the method dampens high-frequency oscillations. This is an intuitive and well-known property of using an exponential moving average (EMA), which is what the s(t) term represents. This is not a deep insight and does not prove why this specific form of smoothing leads to training acceleration.
3. the paper provides no convergence guarantees for its core proposed algorithm.
4. Limited Novelty: The core idea of the paper is to use a weighted average of the current loss and a smoothed version (EMA) of historical loss to determine sampling weights. As noted in the paper's introduction, dynamic sampling methods based on current losses and historical losses  are both well-studied.
5. The motivation of the paper is overly embellished. The core rationale — namely, the need for stability to counteract loss oscillations — naturally leads to EMA, which is already a standard smoothing technique. However, I do not see sufficient justification for the claimed "introduction of first-order loss differences". Equation (3.1) can essentially be viewed as a simple variant of EMA, while Proposition 3.1 mainly concludes that EMA is equivalent to a first-order loss difference. This logical flow is confusing and somewhat circular.
6. The method introduces two key new hyperparameters, betas, in addition to the pruning ratio r and annealing ratios for ESWP. This presents a significant tuning burden for practitioners, making the "plug-and-play" claim difficult to justify.

**Questions:**

See Weaknesses.

---

> ### Author Response · Authors · 2025-11-24
> **Response to Reviewer WM6S**
>
> > **W1**: Convergence analysis of ES
>
> **A1**: We establish the estimate on the  convergence rate of SGD weighted by ES (**Theorem 3.2**). It is shown that under widely-adopted and reasonable conditions (justified in *Remark 1*),  ES-weighted SGD with the constant learning rate  converges, and the convergence rate can be more than sub-linear ($O(1/T)$).
>
> > **W2**: Superiority of ES over EMA
>
> **A2**: With respect, we kindly point out that it is totally an oversimplification to view ES as common EMA.
> - In theory: ES is a non-trivial extension of EMA.
>     - In formulation, ES reduces to EMA by simply setting $\beta_1=\beta_2$.
>     - As shown in **Eq. (3.3)** in **Proposition 3.1**, despite that there are calculations regarding only losses in ES (Eq. (3.1)), Eq. (3.1) also *implicitly* leverages additional **correlations between gradients** to determine sample weights (details in Line 196-206), and this augmented high-order gradient (correlation) information has the coefficient $(\beta_2-\beta_1)$. That is, although both ES and EMA use losses to compute sample weights, the dynamic design of ES enables it to additionally (and implicitly) exploit *gradient* information along the training, while EMA can not. Clearly, different values of two betas also make it possible for ES to flexibly tune the strengths of losses and gradients when computing sample weights.
>     - As discussed in Line 216-219 and Footnote 3 in the original manuscript, ES can take the importance of both current losses and historical weights into account by tuning betas: By setting $(\beta_1, \beta_2) \to (0^+, 1^-)$, we are able to exploit the long-term historical information during training (via $\beta_2$), while simultaneously responding to current losses (via $\beta_1$). This is impossible for EMA.
> - In experiments: The superiority of ES over EMA is consistently verified via both ablation studies and real-world experiments.
>     - As shown in Table 6 (the last two rows) and recapped in Line 461-463 in the original manuscript, ES ("Dif") consistently  outperforms EMA ("NonDif") on both image and language datasets.
>     - As shown in Table 2 and Table 3 in the original manuscript, ES(WP) consistently outperforms UCB ([4], which is exactly a type of  EMA-based sampling) on both small-scale and large-scale tasks.
>
> In Line 170-175, we also update the motivation part regarding the necessity of designing ES as Eq. (3.1). The spirit is, besides the reason to stabilize the weight calculation, it would be also desirable to additionally exploit the higher-order information (like gradient-weighted sampling), with lightweight calculations comparable to loss-weighted sampling.
>
>
> > **W3**: Hyper-parameters tuning
>
> **A3**: We have outlined the hyper-parameters tuning aspect in Section 3.4 in the original manuscript.
> - The primary hyper-parameters are betas in ES. In experiments, we take the default values of $(\beta_1, \beta_2)$, which are obtained by a fine-grained grid search in small-scale simulations (Section 4.3, Paragraph "Choices of $(\beta_1,\beta_2)$"). These defaults are consistently validated to be (locally) optimal in small-scale experiments (Figure 6 and Figure 7), and their superior effectiveness remains in large-scale tasks (Section 4.1, (ii) \& (iii) and Section 4.2). This is *a common routine* of  hyper-parameters tuning, which is also adopted in e.g. [1, 2, 3], to reuse default hyper-parameters (obtained by grid search in small-scale simulations) in large-scale experiments, without further tuning.
> - The other hyper-parameters, including mini-batch sizes, the pruning ratio and annealing epochs, are all responsible for trade-offs between the learning performance and training speed. All of them are user-defined, similar to previous data selection methods such as [1, 3, 4].
>
>
> **References**
>
> [1] Ziheng Qin, Kai Wang, Zangwei Zheng, Jianyang Gu, Xiangyu Peng, Zhaopan Xu, Daquan Zhou, Lei Shang, Baigui Sun, Xuansong Xie, and Yang You. InfoBatch: Lossless training speed up by unbiased dynamic data pruning. In *International Conference on Learning Representations*, 2024.
>
> [2] Ziteng Wang, Jianfei Chen, and Jun Zhu. Efficient backpropagation with variance controlled adaptive sampling. In *International Conference on Learning Representations*, 2024.
>
> [3] Truong Thao Nguyen, Balazs Gerofi, Edgar Josafat Martinez-Noriega, Francois Trahay, and Mohamed Wahib. KAKURENBO: Adaptively hiding samples in deep neural network training. *Advances in Neural Information Processing Systems*, volume 36, pp. 37900–37922, 2023.
>
> [4] Ravi Raju, Kyle Daruwalla, and Mikko Lipasti. Accelerating deep learning with dynamic data pruning. *arXiv preprint arXiv:2111.12621*, 2021.

---

### Author Response · Authors · 2025-11-24
**Global Updates**

We thank all reviewers for their valuable feedback and insightful suggestions. Besides the recognition in comprehensive experiments/ablations with lossless accelerations and light-weighted sampling (Reviewer WM6S \& Reviewer DNpN), reasonable method design with theoretical analysis and validated effectiveness/efficiency on large-scale tasks (Reviewer qPut \& Reviewer evuj), we further update more theoretical results as follows.

1. In **Theorem 3.2**, we establish the estimate on the convergence rate of SGD weighted by ES. It is shown that under widely-adopted and reasonable conditions (justified in *Remark 1*),  ES-weighted SGD with the constant learning rate  converges, and the convergence rate can be more than sub-linear ($O(1/T)$).

2. In **Eq. (3.3)** of **Proposition 3.1**, it is justified that ES can *implicitly* leverage additional **correlations between gradients** along the training to determine sample weights besides the loss re-weighting, despite that there are calculations regarding only losses in ES.

All updates are in blue in the revised paper.

---

### Author Response · Authors · 2025-12-03
**Summary of Rebuttals**

Dear AC,

Thanks for your valuable time to handle our paper. For your convenience, we summarize main reviews and rebuttals here.

This work is about a design of data sampling (re-weighting) algorithm for general training accelerations (called evolved sampling, ES). Its main spirit is to leverage lightweight information like training losses to derive efficient sampling methods. Reviewers recognize this work for the comprehensive experiments/ablations with lossless accelerations achieved by light-weighted sampling scheme (Reviewer WM6S and Reviewer DNpN), reasonable method design with theoretical analysis and validated effectiveness/efficiency on large-scale tasks (Reviewer qPut and Reviewer evuj).

Main concerns from Reviewer WM6S and Reviewer DNpN are concentrated on the moving-average-like design of ES, and no rigorous convergence analysis in the original manuscript. Based on this, we make the following updates during the rebuttal:
1. We add a new convergence theorem (Theorem 3.2). Here, the estimate on the convergence rate of SGD weighted by ES is cleanly established. It is shown that under widely-adopted and reasonable conditions, ES-weighted SGD with the constant learning rate converges, and the convergence rate can be more than sub-linear.
2. We develop original Proposition 3.1 in updated Eq. (3.3). It is justified that ES can *implicitly* leverage additional **correlations between gradients** along the training to determine sample weights besides the loss-based re-weighting, despite of *calculations regarding only losses* in ES.

Based on above and many points in the original manuscript, we clarify that common (exponential) moving average (EMA) is a degenerate case of the proposed ES here, without the following advantages:

1. In theory:
- As shown in Eq. (3.3) in Proposition 3.1, although both ES and EMA use losses to compute sample weights, the dynamic design of ES enables it to additionally (and implicitly) exploit gradient information along the training, while EMA can not.
- As discussed in Line 216-219 and Footnote 3 in the original manuscript, ES can take the importance of both current losses and historical weights into account by tuning hyper-parameters, which is also impossible for EMA.

2. In experiments: The superiority of ES over EMA is consistently verified via both multiple ablation studies and real-world experiments.
- As shown in Table 6 (the last two rows) in the original manuscript, ES ("Dif") consistently outperforms EMA ("NonDif") on both image and language datasets.
- As shown in Table 2 and Table 3 in the original manuscript, ES(WP) consistently outperforms UCB (exact EMA-based sampling) on both small-scale and large-scale tasks.

Therefore, we are very confident that these main concerns have been addressed. We strongly believe that this work proposes a novel, theoretically-grounded, simple and lightweight data re-weighting method, which can be used for general machine learning accelerations in the research direction of data/training efficiency.

Best regards,

Authors

---

### Meta-Review · Area_Chair_m6kJ · 2026-01-05

**Summary:**

The manuscript studies the data sampling (re-weighting) algorithm for general training accelerations. In general, the submission is in good shape.

The reviews reflected a range of perspectives, as detailed in the concerns below. Following a thorough assessment of the reviews and the authors' rebuttals, the AC finds that the manuscript in its current form requires further refinement, particularly regarding its clarity, large-scale experiments, etc.

**Reviewer Concerns:**

Reviewer WM6S argues that
1. Proposition 2.1 analyzes loss-weighted gradient descent, i.e., the baseline "loss" (eq. 2.3), rather than the eq. 3.1.
2. The Fourier analysis in Theorem 3.2 cannot prove why this form of smoothing leads to training acceleration.
3. No convergence guarantees for the proposed algorithm.
4. Limited novelty.
5. The motivation of the paper is overly embellished.
6. The method introduces two key new hyperparameters.

The authors addressed concerns #3, #4, and #6 raised by Reviewer WM6S. After evaluating the rebuttal, the AC believes these responses partially resolve the initial issues.

Reviewer qPut raises concerns about the low baseline performance of ResNet on CIFAR, as shown in Table 2. He/she also questions the code availability. From the view of AC, the authors' response can address the concerns.

Reviewer DNpN argues that (1) the missing comparison with a random sampling baseline, (2) limited novelty, (3) the heuristic design of Eq 3.1, (4) strong assumption in Proposition 2, (5) a detailed justification of the methodology design, (6) hyper-parameter sensitivity, and (7) the validity of proposition 2.1 under standard training regimes.
After reading the rebuttal, AC believes that concerns #1/#3/#4 have been resolved, while concern #2 has been partially resolved, and the answers to concern #5 are not satisfactory.

Reviewer evuj questions the statistical significance, as well as some clarification questions. The AC believes the response is sufficient.

**Reviewer Scores:**

See context above.

---

### Decision · Program_Chairs · 2026-01-26

Reject